# Thermal proteome profiling in bacteria: probing protein state *in vivo*

André Mateus[1] [ID], Jacob Bobonis[1,2], Nils Kurzawa[1,2] [ID], Frank Stein[3], Dominic Helm[3], Johannes Hevler[1], Athanasios Typas[1,*] [ID] & Mikhail M Savitski[1,**] [ID]

## Abstract

Increasing antibiotic resistance urges for new technologies for studying microbes and antimicrobial mechanism of action. We adapted thermal proteome profiling (TPP) to probe the thermostability of *Escherichia coli* proteins *in vivo*. *E. coli* had a more thermostable proteome than human cells, with protein thermostability depending on subcellular location—forming a high-to-low gradient from the cell surface to the cytoplasm. While subunits of protein complexes residing in one compartment melted similarly, protein complexes spanning compartments often had their subunits melting in a location-wise manner. Monitoring the *E. coli* meltome and proteome at different growth phases captured changes in metabolism. Cells lacking TolC, a component of multiple efflux pumps, exhibited major physiological changes, including differential thermostability and levels of its interaction partners, signaling cascades, and periplasmic quality control. Finally, we combined *in vitro* and *in vivo* TPP to identify targets of known antimicrobial drugs and to map their downstream effects. In conclusion, we demonstrate that TPP can be used in bacteria to probe protein complex architecture, metabolic pathways, and intracellular drug target engagement.

**Keywords** *Escherichia coli*; metabolic pathways; protein complexes; target engagement; thermal proteome profiling

**Subject Categories** Genome-Scale & Integrative Biology; Microbiology, Virology & Host Pathogen Interaction; Post-translational Modifications, Proteolysis & Proteomics

**Mol Syst Biol. (2018) 14: e8242**

## Introduction

In the past decade, microbial research has regained traction, with a particular focus on the human microbiome diversity and its importance in health, and on difficult-to-treat infections by multidrug-resistant pathogens. While the microbiome may offer new opportunities for monitoring health and interventions in the long run (Zeller *et al*, 2014; Bullman *et al*, 2017; Maier *et al*, 2018), the shortage of effective antibiotics poses an imminent threat to public health (Brown & Wright, 2016; Tacconelli *et al*, 2018). Discovery of new antibiotics is urgently needed, but developing new drugs is a lengthy, costly, and often unsuccessful process. Interestingly, one of the major bottlenecks at early discovery stages remains the lack of tools for identification of the mode of action (MoA) of antibiotics. At the same time, even for antibiotics used for decades, there is an open debate on how target engagement leads to cell death or arrest (Kohanski *et al*, 2007; Ezraty *et al*, 2013; Keren *et al*, 2013; Liu & Imlay, 2013; Dwyer *et al*, 2014), there is little known about how much off-target effects may play a role in their MoA, and our knowledge on resistance mechanisms is incomplete. On the other hand, studying the physiology of commensal, pathobiont, or pathogenic bacteria is equally important for assessing microbiome-related questions and devising intervention tools.

We have recently introduced a novel technology for detecting protein–drug interactions *in situ* on a proteome-wide scale, termed thermal proteome profiling (TPP; Savitski *et al*, 2014). TPP combines the principle of the cellular thermal shift assay (Martinez Molina *et al*, 2013) with multiplexed quantitative mass spectrometry (MS) using tandem mass tags (TMT; McAlister *et al*, 2012; Werner *et al*, 2012, 2014). In TPP, and in the recently introduced more sensitive 2D-TPP (Becher *et al*, 2016), cells are heated to a range of temperatures and the soluble component of the proteome is interrogated quantitatively at each temperature. Therefore, we can record the melting behavior of thousands of proteins. Protein–drug interactions typically increase the thermal tolerance of proteins, resulting in higher apparent melting points. Thus, comparison of proteome-wide thermostability of drug-treated and drug-untreated cells can lead to identification of drug targets (Savitski *et al*, 2014; Franken *et al*, 2015; Huber *et al*, 2015; Becher *et al*, 2016). In addition to detecting protein–drug interactions, TPP provides a powerful tool for detecting a wide range of physiological changes in protein state: protein–metabolite interactions, post-translational modifications, protein–protein interactions, protein–DNA interactions, and chaperone–client interactions (Savitski *et al*, 2014, 2018; Reinhard *et al*, 2015; Becher *et al*, 2018; Tan *et al*, 2018). Thus, applying TPP at different

1 Genome Biology Unit, European Molecular Biology Laboratory, Heidelberg, Germany
2 Faculty of Biosciences, Heidelberg University, Heidelberg, Germany
3 Proteomics Core Facility, European Molecular Biology Laboratory, Heidelberg, Germany
*Corresponding author. Tel: +49 6221 387 8156; E-mail: typas@embl.de
**Corresponding author. Tel: +49 6221 387 8560; E-mail: mikhail.savitski@embl.de

growth/cell cycle stages, in different nutritional environments, or upon chemical perturbations can yield a unique insight into the cellular physiology and the underlying adaptations taking place (Mateus *et al*, 2017).

We reasoned that adapting TPP to bacteria would hold great promise for deconvoluting the MoA of new compounds with antimicrobial activity and for furthering our understanding of established antibiotics and their downstream effects. More importantly, it would provide a novel and orthogonal way of systematically phenotyping the cell, thereby improving our understanding of basic bacterial biology. Here, we adapt and apply TPP to *Escherichia coli*, and illustrate how it can be used to gain insights into drug–protein interactions, resistance mechanisms, metabolic activity, and protein complex formation.

# Results

### The *Escherichia coli* meltome

To study the melting behavior of the *E. coli* (strain BW25113) proteome, we adapted the TPP protocol (Savitski *et al*, 2014; Franken *et al*, 2015; Reinhard *et al*, 2015) by optimizing the lysis conditions and the temperature range to be compatible with gram-negative bacteria (Fig 1A; see "Materials and Methods" for details). We identified 1,831 proteins (with at least two unique peptides in at least two replicates; Fig EV1A; Dataset EV1), which largely overlapped with recent proteomics datasets obtained from *E. coli* (Wisniewski & Rakus, 2014; Schmidt *et al*, 2016; Fig EV1B). We confirmed that the use of a mild detergent (NP-40) did not affect the solubilization and extraction of membrane proteins—as no bias in quantification was observed when compared with a strong detergent, SDS (Fig EV1C; Dataset EV2).

We calculated apparent melting points ($T_m$) for 1,738 of the identified proteins (Fig 1B). Overall, the *E. coli* proteome was more thermostable than the human proteome ($P < 0.0001$, Mann–Whitney test; Fig 1C), which is consistent with the ability of this organism to grow regularly at temperatures up to 45°C or up to 49°C after evolutionary adaptation (Fotadar *et al*, 2005; Rudolph *et al*, 2010; Blaby *et al*, 2012; Deatherage *et al*, 2017). As previously observed for human proteins (Savitski *et al*, 2014), the $T_m$ correlated very weakly with protein abundance ($r = 0.06$, $P = 0.015$; Fig EV1D) and molecular weight ($r = -0.08$, $P = 0.0009$; Fig EV1E).

Proteins with low $T_m$ included, for example, TypA (BipA) and DeaD, pivotal proteins for growth at low temperatures (Pfennig & Flower, 2001; Charollais *et al*, 2004); GatZ, which has been described to be sensitive to high temperatures (Brinkkotter *et al*, 2002); or the RNA polymerase sigma D factor, RpoD, which is known to lose its function upon heat shock (Blaszczak *et al*, 1995). In addition, multiple essential proteins also melted at low temperatures ($T_m < 52°C$), including topoisomerases (GyrB, ParC, ParE, TopA), proteins involved in DNA replication (DnaA, DnaE) and in cell shape (FtsA, FtsE, FtsI, MrdA, MreB, MukB), as well as multiple components of the small subunit of the ribosome (RpsB, RpsC, RpsD, RpsE, RpsJ, RpsL). In contrast to the overall higher melting temperature of *E. coli* proteins, the small ribosomal subunit melted in a similar temperature range as its human counterpart (Fig 1D). At the opposite side of the spectrum, a number of *E. coli* proteins did not melt in the tested

temperature range (up to 87°C, $n = 93$; Fig EV1I); these were enriched in the Tat translocation system (TatA, TatB, TatE) and outer membrane proteins (e.g., BamC, OmpA, TolC; Dataset EV1).

Protein thermostability increased from the interior to the exterior of the bacterial cell (interquartile range of $T_m$: cytosol (53.3–59°C) < inner membrane (55.5–61.7°C) ≈ periplasm (54.5–61.2°C) < outer membrane (55.7–68.9°C); $P < 0.0001$, Kruskal–Wallis test, with a Dunn's multiple comparison test showing $P < 0.0001$ for all comparisons except inner membrane and periplasm; Figs 1B and E, and EV1H). The most stable periplasmic proteins ($T_m > 65°C$) included all the superoxide dismutases (SodA, SodB, SodC), chaperones (FkpA, Skp, Spy, DsbC, HdeA, HdeB), and ligand-binding subunits of ATP-binding cassette (ABC) transporters (such as, PotD, PstS, MlaD, LolA, DppA, or ArtI), all engaged in cargo or ligand binding, suggesting that TPP may capture active periplasmic processes. In the outer membrane, integral membrane proteins were generally more thermostable than lipoproteins ($P = 0.031$, Mann–Whitney test; Fig EV1G). Outer membrane porins (OMPs) are known to be particularly stable once assembled and even resistant to denaturation (Ureta *et al*, 2007; Burgess *et al*, 2008; Stanley & Fleming, 2008; Roman & Gonzalez Flecha, 2014). Compatible with this, we observed that multiple OMPs showed a biphasic melting behavior, with a fraction melting at lower temperature and another fraction being stable up to 87°C (Fig 1B). A fraction of the proteins in the outer membrane (mostly integral proteins) showed an increase in solubility at approximately 55°C (Fig EV1G), which could be linked to a reported disorganization of the outer membrane at this temperature (Tsuchido *et al*, 1985).

To compare our results to a complementary approach probing protein thermal unfolding based on limited proteolysis (LiP-MS; Leuenberger *et al*, 2017), we further determined the $T_m$ of proteins in an *E. coli* lysate (by lysing the cells prior to heat treatment). Despite a good agreement between our lysate and living cell experiments ($r = 0.82$, $P < 0.0001$; Fig EV1F; Dataset EV3), we observed only a moderate correlation with the limited proteolysis approach ($r = 0.45$, $P < 0.0001$; Fig 1F); particularly, the ribosome was considerably less thermostable in our experiments. Since ribosomal thermostability is affected by the presence of magnesium ions (Friedman *et al*, 1967; Piazza *et al*, 2018), we repeated our lysate experiments with a similar concentration of $MgCl_2$ (10 mM) to that described in Leuenberger *et al* (2017). This resulted in a much improved correlation ($r = 0.65$, $P < 0.0001$; Fig 1G; Dataset EV3), due to the overall increased stability of the ribosome. Interestingly, RplA, RplJ, RplL, and RpsA were not affected by increased magnesium concentration. These are structurally distinct from the other ribosomal proteins, as they are part of the lateral stalk (RplJ and RplL; Choi *et al*, 2015), the L1 stalk (RplA; Reblova *et al*, 2012), or known not to always be associated with the 30S subunit (RpsA; Duval *et al*, 2013). The TPP data showed on average higher $T_m$ than the LiP-MS approach, which probably reflects the differences between the two approaches. These include technical differences—a longer heat-treatment time in LiP-MS that might lead to more extensive unfolding—and conceptual differences—LiP-MS detects local events in protein unfolding, while TPP studies protein aggregation that might require the unfolding of a substantial fraction of the protein.

In summary, the *E. coli* proteome is more thermostable than the human one, thermostability of proteins correlates with their subcellular localization, and proteins with low temperature-related

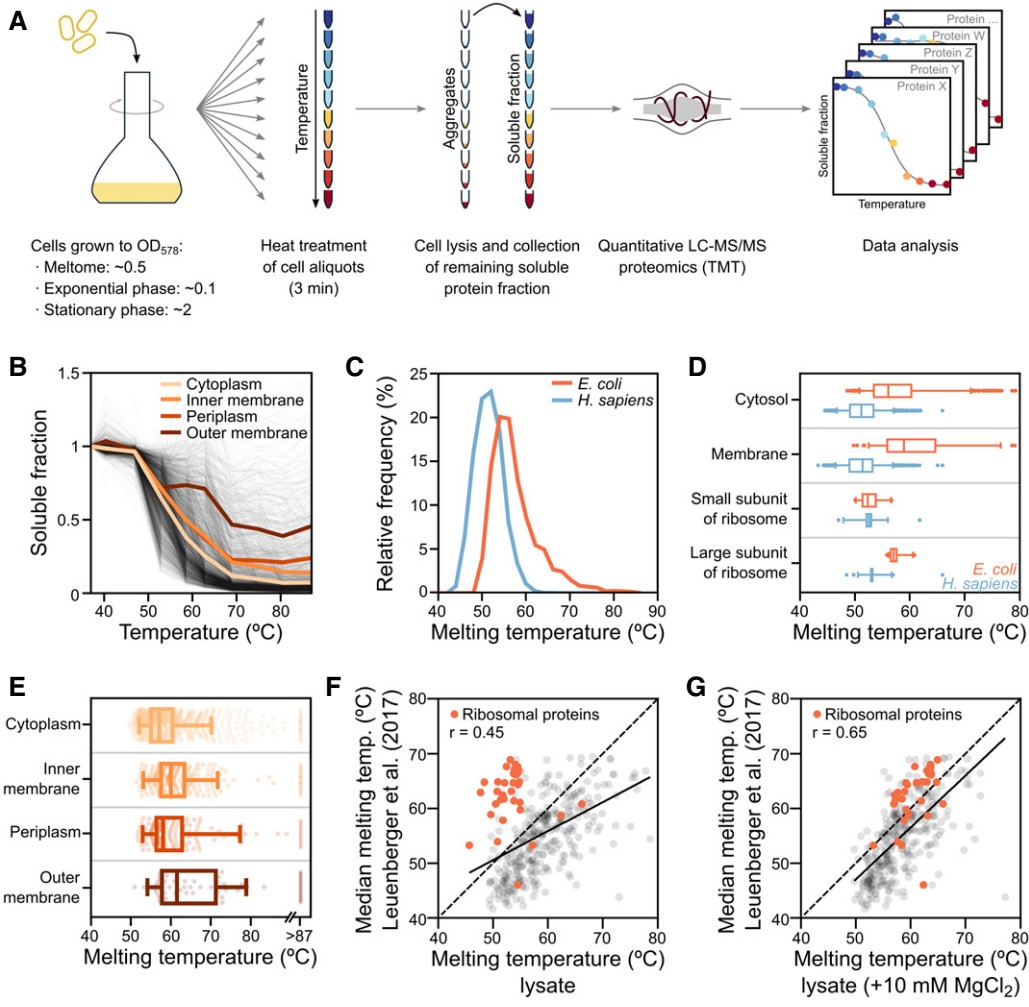

**Figure 1. Thermal proteome profiling in *Escherichia coli*.**

A    Thermal proteome profiling protocol overview. After cells are grown to a specified optical density ($OD_{578}$), aliquots are heated to a range of temperatures, lysed, and the remaining soluble fraction of the proteome is collected. Mass spectrometry-based proteomics (using tandem mass tags, TMT) is then used to quantify the amount of protein at each condition, and melting curves are plotted for each protein.

B    Melting curves for *E. coli* proteins. The average melting curve for each cellular compartment is shown.

C    Distribution of melting temperatures ($T_m$) of the *E. coli* and the human proteomes.

D    Distribution of melting temperatures ($T_m$) of the *E. coli* and the human proteomes according to selected gene ontology terms. Line represents the median, box represents the interquartile range, and whiskers of the box plots represent the 5th and 95th percentiles.

E    Distribution of melting temperatures ($T_m$) of the *E. coli* proteome according to their cellular compartment. Box plots are plotted as panel (D).

F    Correlation of melting points in lysate determined by TPP (this study) with melting points determined by limited proteolysis coupled to mass spectrometry (Leuenberger *et al*, 2017). For the results from Leuenberger *et al* (2017), the median melting point of the reported peptides for each protein was used. Only proteins with at least two identified peptides were compared. Red dots represent ribosomal proteins, which generally appear less thermostable in TPP.

G    Correlation of melting points in lysate determined by TPP upon addition of 10 mM $MgCl_2$ (this study) with melting points determined by limited proteolysis coupled to mass spectrometry (Leuenberger *et al*, 2017). For the results from Leuenberger *et al* (2017), the median melting point of the reported peptides for each protein was used. Only proteins with at least two identified peptides were compared. Red dots represent ribosomal proteins.

functions, as well as several essential proteins, have low thermostability.

## Impact of growth phase on protein thermostability and abundance

We tested the impact of growth phase on the thermostability of the proteome, since we expected differences in protein activity to be reflected in thermostability (Savitski *et al*, 2014; Reinhard *et al*, 2015)—for example, proteins might be stabilized by substrates (indicating flux through the pathway), products (indicating inhibition), or allosteric regulators. For this, we harvested cells growing in LB medium in exponential phase ($OD_{578} \approx 0.1$) and in the transition to stationary phase ($OD_{578} \approx 2$; Fig 1A). We identified 39 proteins with significant differences in thermostability ($P < 0.001$, non-parametric analysis of response curves (see "Materials and

Methods"; Fig 2A; Dataset EV4). Among these were lactate dehydrogenase (LldD), and some of the members of the respiratory complex I (NuoA and NuoL) and II (SdhB and SdhD), which were stabilized in the transition to stationary phase (all the other members of these complexes were also stabilized, albeit not significantly). This is consistent with stronger respiratory activity in stationary phase, as observed by the higher levels of triphenylformazan formed from the reduction in triphenyltetrazolium chloride ($P = 0.0027$, Student's $t$-test; Fig 2C). Accordingly, the glycerol kinase (GlpK), which converts glycerol to $sn$-glycerol 3-phosphate, and the $sn$-glycerol 3-phosphate uptake transporter (GlpT) were also stabilized at this growth phase; glycerol can only be used through respiration. Furthermore, YjiY, a novel specific transporter of pyruvate, which is induced exactly in the transition to stationary phase to import pyruvate (Kristoficova $et$ $al$, 2017), was also more thermostable at this phase. Conversely, the peptide transporters, MppA and OppA, and tryptophanase (TnaA) were destabilized in the transition to stationary phase, suggesting that the cell had exhausted small peptides and had started producing indole. TnaA has been shown to form a single inactive focus in the cell pole during exponential phase, but disperse and gain activity in stationary phase (Li & Young, 2015). The decrease in $T_m$ is presumably reflective of the shift from the focal conformation to free active tetramers (Li & Young, 2015).

In addition to protein thermostability, we could estimate protein abundances at the two growth phases—as traditional proteomics approaches (Aebersold & Mann, 2016)—by comparing the intensities of the signal at the lowest temperature (37°C), since the vast majority of proteins have not yet started to melt (Fig 1B). We found 17 proteins exclusively expressed in exponential phase (identified in at least two replicates by at least three unique peptides, while not being detectable in three replicates of stationary phase) and 57 proteins only expressed in the transition to stationary phase. In addition, five proteins were down-regulated in the transition to stationary phase and 53 were up-regulated [$P < 0.01$, using a linear model to assess differential expression (see "Materials and Methods"); Fig 2B; Dataset EV4]. Proteins that were absent in this growth phase included the iron (III) hydroxamate transporter (FhuC, FhuD).

Proteins with exclusive or higher abundance included the sigma factor S (RpoS)—note that its regulon is activated later in stationary phase (Weber $et$ $al$, 2005)—and multiple proteins involved in metabolic processes, the majority of them being regulated by cAMP receptor protein (CRP)—known to be dominant at this growth stage in rich media. These included proteins mediating fatty acid beta-oxidation (FadA, FadI, FadJ); the transport of maltose (MalF, MalK) or sorbitol (SrlA, SrlB, SrlD, and SrlE); and the catabolism of threonine (TdcB, TdcE), glycerol (GlpB, GlpC), glucarate (GarL, GudD), N-acetylneuraminate (NanE, NanK), and glycine (GcvP, GcvT). Members of respiratory complexes and proteins involved in glycerol, lactate, pyruvate, and tryptophan metabolism had affected not only their thermostability (as described above), but also their abundance.

In summary, we observe thermostability changes for many metabolic enzymes and complexes during the transition to stationary phase, many of which are consistent with changes in their protein activity, and the cell using less efficient energy sources and slowing down growth at this stage. For most of these proteins, the increase in thermostability coincided with increase in abundance. The only prominent outlier, TnaA, which had lower thermostability, is also known to have higher activity at this growth phase (Li & Young, 2015).

## Melting behavior of protein complexes

Next, we evaluated the thermostability behavior of protein complexes. We expected that proteins of the same complex would co-melt, since the melting of one of the components can destabilize the remaining complex members—causing them to have similar melting behavior, a phenomenon recently called thermal proximity coaggregation, TPCA (Tan $et$ $al$, 2018). To test this, we calculated the average Euclidean distance between melting curves of proteins of the same complex in $E.$ $coli$ and in human cells for comparison (see "Materials and Methods"; Dataset EV5). A large proportion of protein complexes melted coherently for both human and $E.$ $coli$ protein complexes (Fig 3A; Dataset EV6), but the bacterial complexes had a more variable melting behavior (i.e., higher Euclidean distance).

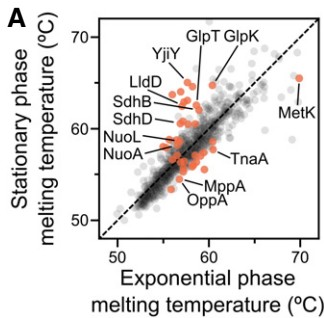 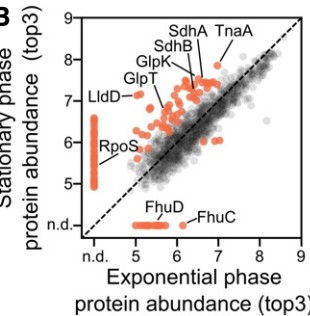 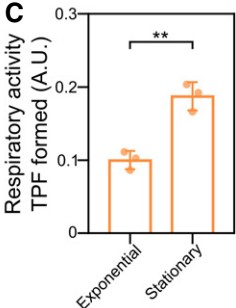

**Figure 2.  Impact of growth phase on the *Escherichia coli* meltome and proteome.**

A   Melting temperatures ($T_m$) of proteins in exponential and transition to stationary growth phases. Proteins highlighted in orange indicate significantly different melting behavior.

B   Protein abundance in exponential and transition to stationary growth phases, as measured by the top3 intensity corresponding to the lowest temperature (see "Materials and Methods"). Proteins highlighted in orange indicate significantly different levels. Proteins were considered not detectable (n.d.) in one condition, if absent in three replicates in that condition, but detectable by at least three unique peptides in at least two replicates in the other condition.

C   Respiratory activity in exponential and stationary cells determined as the conversion of triphenyltetrazolium chloride to triphenylformazan during the same time and normalized by OD (~number of cells). $n = 3$; error bars represent standard deviation; **$P < 0.01$, Student's $t$-test.

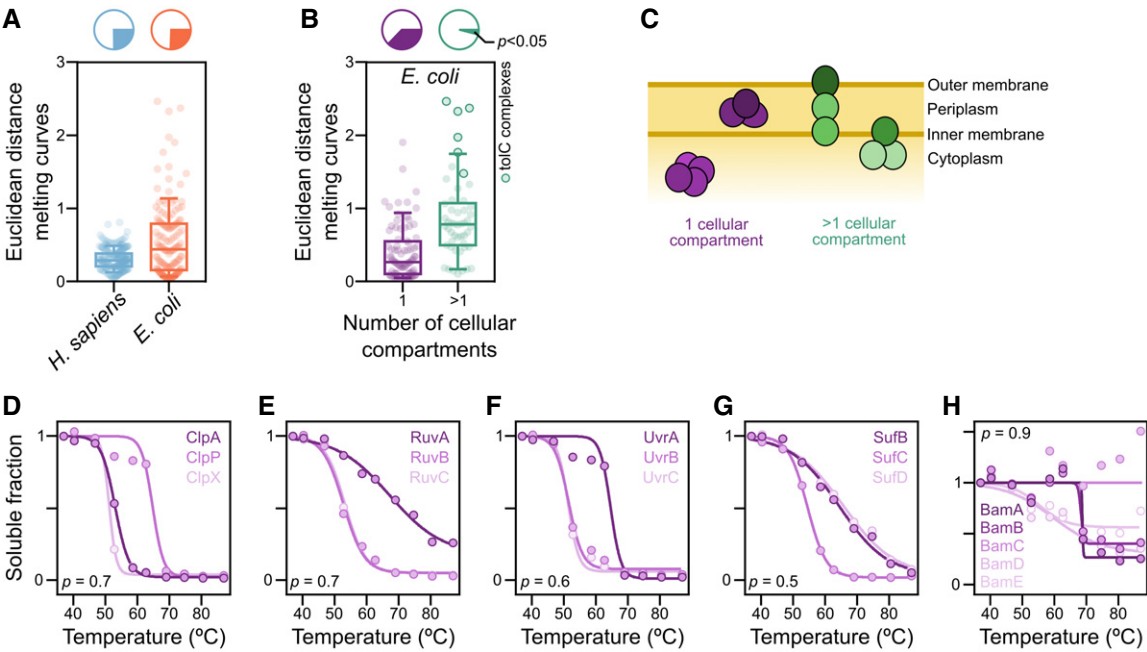

**Figure 3.  Melting behavior of protein complexes.**

A    Melting behavior of protein complexes from human and *Escherichia coli* was measured by the average Euclidean distance between the melting curves of proteins from each complex. Line represents the median, box represents the interquartile range, and whiskers of the box plots represent the 10[th] and 90[th] percentiles. Pie charts represent the fraction of protein complexes that melt coherently (compared with a distribution of 10,000 random complexes; $P < 0.05$).

B    Comparison of the melting behavior of protein complexes located in a single cellular compartment or in multiple compartments. Line represents the median, box represents the interquartile range, and whiskers of the box plots represent the 10[th] and 90[th] percentiles. Pie charts represent the fraction of protein complexes that melt coherently (compared with a distribution of 10,000 random complexes; $P < 0.05$).

C    Schematic representation of complexes located in a single cellular compartment or in multiple compartments.

D–H    Melting curves for examples of non-co-melting complexes located in the same cellular compartment: (D) ClpP protease complex, (E) Ruv DNA repair complex, (F) Uvr DNA repair complex, (G) Suf Fe-S biogenesis complex, and (H) Bam outer membrane porin assembly complex. *P* indicates the probability that the complex melts coherently (compared with a distribution of 10,000 random complexes).

Interestingly, most of the non-co-melting bacterial complexes spanned different cellular compartments (Fig 3B and C), as for example, multidrug efflux complexes that are composed of inner membrane, periplasmic, and outer membrane components. This suggests that protein localization overrides the effect of stabilization by other complex members and that presumably many of these across-compartment complexes consist of more stable co-localized subcomplexes. For example, the inner-membrane-bound subunits of efflux pumps melted very similarly indicating that they form separate subcomplexes, which is consistent with the differential localization and dynamics of AcrB in the presence or absence of TolC (Bergmiller *et al*, 2017). Nevertheless, such protein complexes still have the thermostability of their members linked, as evidenced by the destabilization of all inner membrane complex components upon removing their outer membrane counterpart, TolC (see below "Effect of knocking out *tolC* on proteome thermostability and abundance").

Complexes in the same compartment with non-co-melting behavior could generally be attributed to the formation of previously observed subcomplexes. For example, these included the following: proteases, in which the catalytic subunit (ClpP or HslV) was more thermostable than the also singly acting chaperonin (ClpA, ClpX, or HslU; Rohrwild *et al*, 1996; Ortega *et al*, 2004; Fig 3D); DNA repair complexes (Ruv and Uvr), in which the respective more thermostable DNA binding components (RuvA and UvrA) can also act

on their own (Van Houten *et al*, 1987; Shiba *et al*, 1991; Fig 3E and F); the Suf complex, in which SufBD is known to form the core of the complex (Hirabayashi *et al*, 2015) and was more thermostable than SufC (Fig 3G); and the Bam complex, the machinery for outer membrane porin insertion, in which three subcomplexes were apparent (BamAB, BamDE, and BamC; Fig 3H; Noinaj *et al*, 2017). BamC was completely thermostable, which is consistent with its enigmatic topology, i.e., high flexibility in crystal structures and appearing surface-exposed in *in vivo* experiments (Bakelar *et al*, 2016; Gu *et al*, 2016; Han *et al*, 2016; Iadanza *et al*, 2016; Noinaj *et al*, 2017). An overview of the co-melting behavior of all *E. coli* protein complexes detected is available in Fig EV2.

Taken together, we observe that although complex architecture affects the coherence of complex melting, spatial location of proteins has a stronger effect on melting behavior than complex membership.

### Effect of knocking out *tolC* on proteome thermostability and abundance

To further evaluate whether proteins are indeed co-stabilized if present in a complex, we studied the cellular impact of removing one of the members of a complex. For this, we compared a *tolC* knockout strain (Δ*tolC*) to the wild-type *E. coli* (WT) strain using two-dimensional TPP (2D-TPP; Fig 4A; Becher *et al*, 2016, 2018).

TolC is a promiscuous member of efflux pump complexes that span multiple cellular compartments and are required for pumping out toxic (xeno-)metabolites, including a number of antibiotics, from the cell. 2D-TPP allowed us to assess the impact of Δ*tolC* on the abundance (abundance score) and thermostability (stability score) of each protein of the proteome (see "Materials and Methods"; Dataset EV7). Overall, the abundance of 38 proteins was

significantly affected (11 proteins were down-regulated, and 27 proteins were up-regulated; Figs 4B and EV3A), while 55 proteins showed differences in thermostability (17 destabilized and 38 stabilized; Figs 4C and EV3B).

Remarkably, despite the disparate melting behavior of TolC compared to its known complex members (Zgurskaya *et al*, 2011), these proteins were significantly destabilized (AcrAB, MdtEF, EmrA,

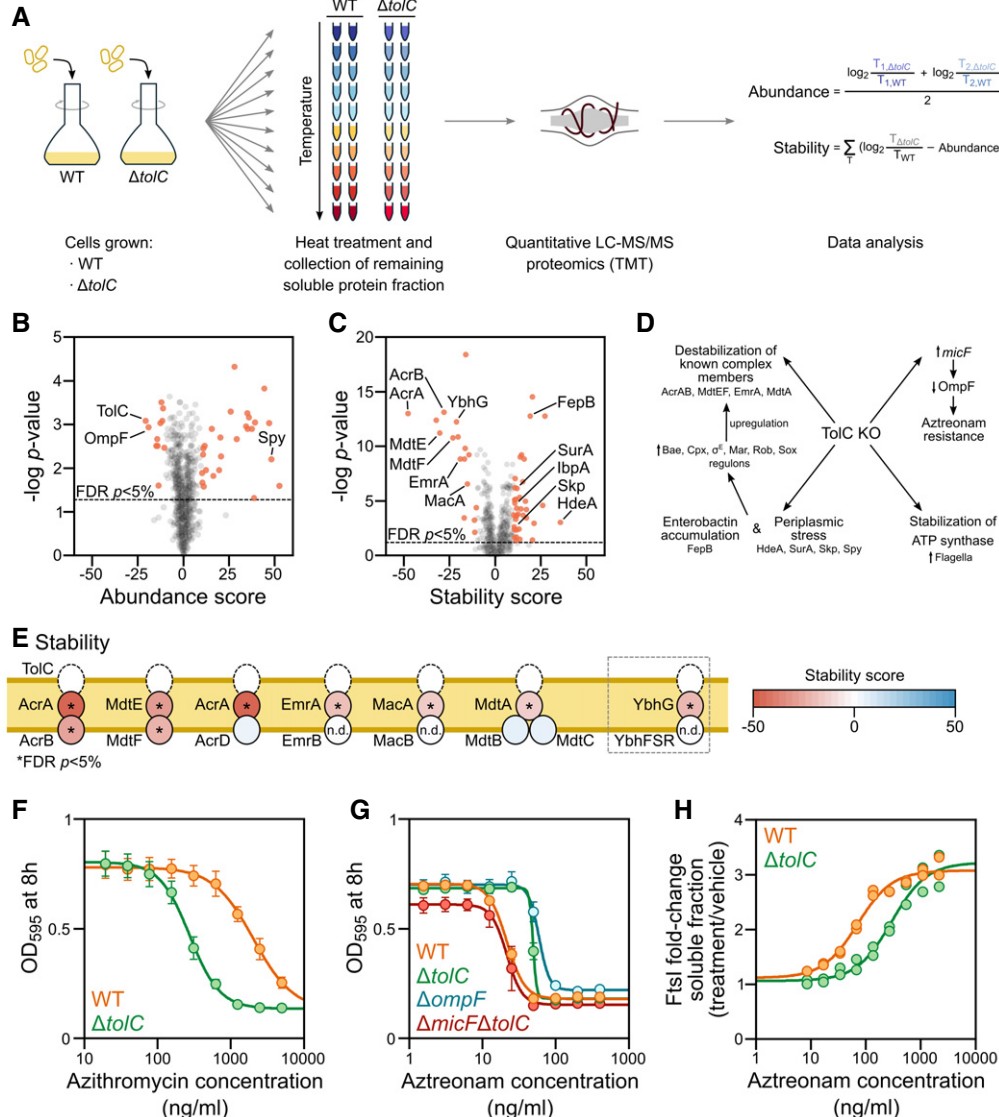

**Figure 4. Effect of gene knockout on protein thermostability and abundance.**

A Two-dimensional thermal proteome profiling (2D-TPP) protocol overview. Wild-type (WT) and *tolC* knockout strain (Δ*tolC*) were grown and prepared in a similar manner to what is described in Fig 1A. For each protein, abundance and stability scores were calculated.

B, C Volcano plots for abundance (B) and stability (C) scores for each identified protein in Δ*tolC* compared to WT (TolC signal is detected at noise level in the Δ*tolC* strain, due to its presence in WT and TMT quantification rarely producing missing values). Proteins highlighted in orange show significant changes [false discovery rate (FDR) *P* < 0.05 and absolute score > 10].

D Proposed mechanism for abundance and stability hits of Δ*tolC*.

E Schematic representation of TolC complexes and the stability scores of their members. YbhG is a member of a putative efflux transporter. *False discovery rate (FDR) *P* < 0.05 and absolute score > 10. n.d. not detected.

F, G Cell growth (as measured by OD595) after 8 h in the presence of azithromycin (F) or aztreonam (G) in WT, Δ*tolC*, Δ*ompF*, and Δ*micFΔtolC* cells (*n* = 4; error bars represent standard deviation).

H Target engagement affinity of aztreonam in WT and Δ*tolC* cells, measured by thermal proteome profiling–compound concentration range (TPP-CCR). Stabilization of the main known target of aztreonam (FtsI) is shown.

MacA, and MdtA; Figs 4E and EV3B). Some known members were not detected (e.g., EmrB and MacB), and others were not affected (AcrD and MdtBC)—likely because they are not assembled under growth in LB medium (Yamamoto *et al*, 2016). Apart from the known interactors of TolC, YbhG—a gene of unknown function, but belonging to an operon encoding a putative ATP-binding cassette transporter (Yamanaka *et al*, 2016), and recently found to be affinity purified with TolC (Babu *et al*, 2017)—was destabilized and showed a similar increase in protein abundance as other TolC complex members (Fig 4E). This suggests that YbhGFSR forms yet another efflux pump complex with TolC. Other destabilized proteins included members of the glycerol-3-phosphate dehydrogenase (GlpA and GlpB) and multiple OMPs (OmpC, OmpX, FhuA, and LamB). Another abundant OMP, OmpF, was the most down-regulated protein in Δ*tolC* cells, suggesting that the cell is trying to reduce levels and activity of OMPs required for the import of external molecules, when its major efflux pumps are not working (Misra & Reeves, 1987).

Conversely, we observed stabilization of FepB, a periplasmic protein involved in iron uptake via the siderophore enterobactin. Enterobactin is synthesized in the cytoplasm and secreted to the extracellular milieu via TolC (Bleuel *et al*, 2005). In the Δ*tolC* background, enterobactin accumulates in the periplasm, where it is toxic. FepB, which usually binds the iron-conjugated form, can also bind the apo-form of enterobactin to remove it from this compartment and bring it back to the cytoplasm to alleviate its toxicity (Vega & Young, 2014). A number of additional proteins involved in protein folding and quality control in the periplasm (SurA, Skp, HdeA, DsbG) were also stabilized. To validate that increased thermostability was due to increased activity—i.e., engagement of chaperones with unfolded proteins in the envelope—we assessed the importance of SurA, one of the two major chaperone pathways for ushering unfolded outer membrane β-barrel proteins through the periplasm (Goemans *et al*, 2014), in the absence of *tolC*. Although Δ*surA* and Δ*tolC* single mutants are viable and have minimal fitness costs, we could not obtain the double knockout (Δ*surA*Δ*tolC*) in lysogeny agar (LA) selecting for kanamycin-resistant colonies. We hypothesized that this was caused by the low levels of divalent cations in LA (Papp-Wallace & Maguire, 2008), which are important for the structural integrity of the lipopolysaccharides present in the outer membrane (van Alphen *et al*, 1978). In agreement, we could obtain the Δ*surA*Δ*tolC* in cation-adjusted M9 minimal medium. This suggests that the integrity of the outer membrane is severely compromised in this mutant, as evidenced by the higher drug sensitivity in low divalent cationic concentrations (due to higher permeability to kanamycin; Fig EV3C) and by the hypersensitivity to salt (NaCl; Fig EV3D and E). Consistently with the Δ*tolC* mutants cells experiencing periplasmic stress (and thus periplasmic chaperones being more active), we observed higher levels of a number of members of the envelope surveillance systems, Cpx, Bae, and σ$^E$ (Fig 4C and D), and the Mar/Rob/Sox regulon, in agreement with previous results (Corbalan *et al*, 2010; Rosner & Martin, 2013)—note that σ$^E$ system senses unfolded β-barrel proteins in periplasm (Mecsas *et al*, 1993; Walsh *et al*, 2003). In addition, this stress response also provided a feedback on the efflux pump expression (Fig 4C and D; Rosner & Martin, 2013). Furthermore, a strong increase in the abundance of flagella subunits and a concomitant stabilization of ATP-synthase subunits were in agreement with

previous reports on increased motility of AcrAB-TolC pump mutants in *E. coli* (Ruiz & Levy, 2014). Since both flagella rotation and AcrAB-TolC are proton-motive force driven (Paul *et al*, 2008; Nagano & Nikaido, 2009), the absence of TolC may allow for more proton gradient to be used by both the ATP synthase and flagella. Finally, multiple members of different metabolic pathways were stabilized in Δ*tolC* (e.g., CysP, FabD, MaeA, PyrF, or SrlB), which is consistent with the global metabolic changes caused by lower efflux pump activity in *E. coli* (Zampieri *et al*, 2017a).

We next asked whether we could explain the net impact of knocking out *tolC* on antibiotic sensitivity, based on the complex cellular changes we observed. Deleting *tolC* increases the sensitivity to numerous antibiotics (Zgurskaya *et al*, 2011)—since antibiotics are not effluxed and accumulate inside the cell (as can be seen for azithromycin; Fig 4F). However, high-throughput screens have revealed opposite trends for a few antibiotics, such as aztreonam, which is less potent in the Δ*tolC* background (Nichols *et al*, 2011). We confirmed that the Δ*tolC* mutant is more resistant to aztreonam (Fig 4G). Moreover, we used TPP to show that this was likely due to lower intracellular concentrations of aztreonam, as the drug engaged its known targets (FtsI and MrcA) with a lower apparent affinity in the Δ*tolC* strain than in the wild type (Figs 4H and EV3F; Dataset EV8). The decreased intracellular concentration of a drug in a cellular background lacking a major efflux pump seems counterintuitive, but could be rationalized by a decreased uptake due to OmpF down-regulation. Indeed, there are prior indications that OmpF could be responsible for aztreonam uptake (Hiraoka *et al*, 1989; Nishino *et al*, 2003), and we confirmed this in the Δ*ompF* strain (Figs 4G and EV3D). OmpF down-regulation in the Δ*tolC* mutant is caused by an increase in *micF* (Misra & Reeves, 1987; Fig 4D). Therefore, in a Δ*tolC*Δ*micF* background, in which OmpF levels are similar to wild type, the potency of aztreonam was similar to that in wild-type cells (Figs 4G and EV3E).

In summary, deletion of *tolC* leads to destabilization of multiple efflux pumps and to marked periplasmic stress and metabolic changes (Fig 4D). One of the ways the cell tries to deal with the stress is by altering the levels of OMPs. Decreased OmpF levels result in an unexpected increased resistance to aztreonam for the generally antibiotic hypersensitive Δ*tolC* mutant. More broadly, these results illustrate how TPP can be used to infer biological insights into the altered cell physiology of mutants—in this case a mutant of major antibiotic resistance determinant in enterobacteria. Notably, many of these changes in cellular physiology, such as the destabilization of efflux pump complexes, the periplasmic quality control response, and the failure to export enterobactin, would be impervious to any other genome-wide phenotyping approach.

### Antimicrobial target identification and effects on cell physiology: ampicillin and ciprofloxacin

We also used 2D-TPP to identify the direct targets and the mode of action of two well-known antibiotics, one acting in the periplasm and one in the cytoplasm of bacterial cells (Fig 5A).

We first incubated living cells with ampicillin, an antibiotic inhibiting peptidoglycan biosynthesis by binding to multiple penicillin-binding proteins (PBPs; Lewis, 2013). This resulted in 130 proteins being significantly affected in their thermostability and/or abundance. Among these, MrcA (PBP1A), FtsI (PBP3), DacB

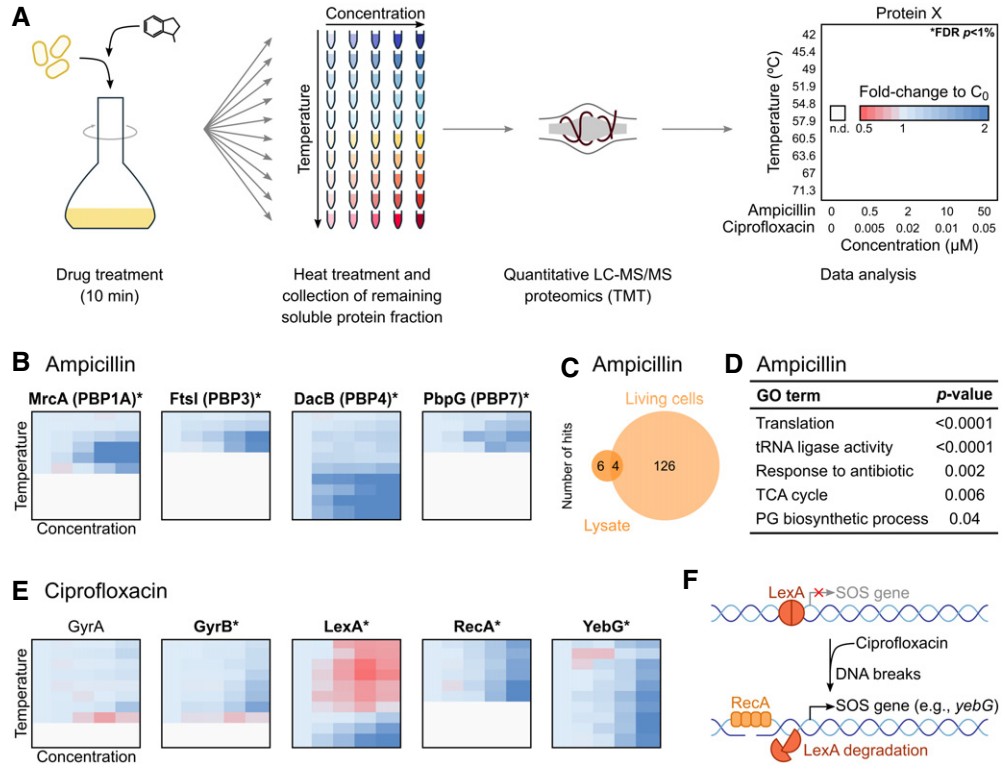

**Figure 5. Target identification of ampicillin and ciprofloxacin.**

A   2D-TPP protocol overview. After treatment with different concentrations of antibiotics, the cells were prepared in a similar manner to what is described in Fig 1A. For each protein and temperature, the signal intensity was normalized to the vehicle control.

B   Heatmaps for targets of ampicillin in living cells, with coloring according to panel (A). *FDR controlled at 1% using a bootstrapped permutation approach.

C   Number of stabilized or destabilized proteins in lysate and living cells after treatment with ampicillin.

D   Example of top gene ontology terms enriched in proteins affected in living cells after treatment with ampicillin.

E   Heatmaps for targets of ciprofloxacin in living cells, with coloring according to what is described in panel (A). *FDR controlled at 1% using a bootstrapped permutation approach.

F   Schematic representation of SOS response. LexA binds to promoters of SOS response genes and represses their transcription. Treatment with ciprofloxacin induces single- and double-stranded DNA breaks that recruit RecA to DNA, causing auto-cleavage of LexA and expression of SOS response genes (e.g., *yebG*; Andersson & Hughes, 2014).

(PBP4), and PbpG (PBP7) were significantly stabilized (Fig 5B; Dataset EV9). We calculated an $IC_{50}$ for these proteins, which matched previous reports (Kocaoglu & Carlson, 2015), with the highest affinity being toward DacB, followed by FtsI and PbpG at similar levels (Dataset EV9). To confirm that these were the direct targets of this compound, we performed 2D-TPP in cell lysate, in which, due to disruption and dilution of cellular contents, downstream effects are not present. In lysate, only 10 proteins were significantly affected, with MrcA, and DacB being thermally stabilized (Figs 5C and EV4A; Dataset EV10). FtsI and PbpG were below the limit of detection for most temperatures and, for that reason, their thermostability could not be assessed.

Apart from PBPs, AmpC—a β-lactamase that mediates resistance to ampicillin (Jacoby, 2009)—was the only other protein stabilized both in living cells and lysate. In living cells, the downstream affected proteins were involved not only in cell-wall-related processes (AmiA, MltA, MltB, MppA, FtsZ, MinD, and MreB), but also in translation (multiple subunits of the ribosome), tRNA biosynthesis (multiple tRNA ligases), protein quality control (e.g., ClpB, ClpX, Lon, and Tig), and central metabolism [multiple

enzymes involved in glycolysis and the tricarboxylic acid cycle, in agreement with previous observations from metabolomics experiments (Zampieri *et al*, 2017b)] (Fig 5D).

Finally, we incubated cells with ciprofloxacin—an antibiotic that impairs DNA replication by inhibiting DNA gyrase (GyrAB) and topoisomerase (ParCE; Lewis, 2013). The known targets of this compound were stabilized in living cells (though only GyrB stabilization was significant; Fig 5E; Dataset EV11), but not in the lysate (Fig EV4B; Dataset EV12)—consistent with ciprofloxacin only binding gyrase in the presence of DNA (Critchlow & Maxwell, 1996), which was digested during lysate preparation. Furthermore, a strong SOS response, the hallmark of DNA damaging agents, was apparent in living cells (Andersson & Hughes, 2014), namely the degradation of LexA (Fig 5E, visible at the lowest temperatures), the thermostabilization of RecA, and the up-regulation of SOS genes, such as YebG (Fig 5F). Moreover, a similar effect to ampicillin on ribosomes and tRNA ligases was observed, in addition to a strong destabilization of histone-like proteins (H-NS, HU), which could be a direct consequence of changing the DNA-supercoiling by DNA gyrase and topoisomerase inhibition.

Overall, our results indicate that TPP not only captures the direct targets of antimicrobial compounds, but also can be used to monitor determinants of resistance and downstream consequences on cellular physiology. Although not yet conclusive, our results are indicative of previously unknown general effects on protein translation and quality control upon exposure to antibiotics, and a rather diverse stress alarm response of the cell to those cytotoxic agents.

## Discussion

We have analyzed the *in vivo* melting behavior of the *E. coli* proteome and discovered that thermostability is influenced by protein subcellular location, structure, activity state, and drug binding. The information about protein thermostability is complementary to protein abundance (the latter, traditionally obtained in proteomics analyses) and can be used to further the understanding of bacterial cell physiology, drug mode of action, and drug resistance mechanisms.

In agreement with the ability of *E. coli* to divide at temperatures up to 45°C (Fotadar *et al*, 2005; Rudolph *et al*, 2010; Blaby *et al*, 2012; Li & Ganzle, 2016; Deatherage *et al*, 2017), all identified proteins were resistant to heat-induced aggregation up to this temperature (the lowest $T_m$ was 48.6°C for GatB). Contrary to previous suggestions based on *in vitro* experiments (Dill *et al*, 2011), we did not observe the denaturation of the whole proteome at this temperature. However, as proposed by a recent publication using a limited proteolysis strategy to assess protein thermostability in lysates (Leuenberger *et al*, 2017), we observed multiple essential genes coding for proteins with low $T_m$ that might constrain growth at higher temperatures. Among the essential proteins with low $T_m$, we identified MrdA, MreB, RpoD, RpsD, and RpsJ (all of which with $T_m < 52$°C), which have also been shown to be mutated in *E. coli* experimentally evolved to grow at higher temperatures (Blaby *et al*, 2012; Deatherage *et al*, 2017). Future studies, probing by TPP closely related organisms with different heat tolerance, may pinpoint the exact bottlenecks for growth at higher temperatures and whether those are conserved across species. Importantly, TPP is performed *in situ* and can also capture epistatic effects, such as the impact of protein quality control on the thermostability of other proteins (Weibezahn *et al*, 2004; Rene & Alix, 2011; Lee *et al*, 2018). The similar $T_m$ (in *E. coli* and humans) of the members of the small subunit of the ribosome, the most conserved genes across all kingdoms of life (Isenbarger *et al*, 2008), indicates that not only sequence and structure are conserved, but also melting behavior. In addition, heat-induced denaturation can be an evolutionary strategy for deactivating proteins (possibly to reduce fitness costs), as shown by the low $T_m$ of proteins that are only required for growth at low temperature. For example, DeaD ($T_m = 49.9$°C) and SrmB ($T_m = 51.1$°C), both ATP-consuming RNA helicases required during cold shock, can be removed at higher temperatures when they are not required (Charollais *et al*, 2003; Turner *et al*, 2007).

The strong resistance to heat- or chemical-induced denaturation of proteins in the outer membrane, particularly once folded, has been previously linked to their structure (β-barrel; Ureta *et al*, 2007; Burgess *et al*, 2008; Stanley & Fleming, 2008; Roman & Gonzalez Flecha, 2014). The melting profile of these proteins suggests that each of them exists as two populations—one with a low $T_m$ (marked by an initial decrease in the amount of remaining soluble protein and corresponding to an unfolded state), and one with a high $T_m$ (marked by a plateau after the initial decrease and corresponding to a folded state) —with the observed melting curve reflecting the sum of these two populations. Mathematical modeling can theoretically determine the proportions of each of these populations (Becher *et al*, 2018), and TPP could be used in the future to study global effects on the mechanisms of *in vivo* protein assembly in the outer membrane, for example, by studying mutants with impaired protein folding and systematically determining the proportions of properly folded protein.

Protein thermostability and abundance changes observed in the transition to stationary phase are consistent with the changes at the metabolome and transcriptome levels (Jozefczuk *et al*, 2010), such as the increase in respiration activity (Zambrano & Kolter, 1993; Orman & Brynildsen, 2015), the complete consumption of certain nutrients from the media and the shift to different energy sources (Kristoficova *et al*, 2017), and the production of indole (Li & Young, 2015). Moreover, the concurrent increase in abundance for the majority of the stabilized proteins indicates that the cell reacts to higher metabolic fluxes by increasing protein expression (Noor *et al*, 2016). For the periplasmic binding component of ABC transporters, thermostability seems to be a strong indicator of activity and ligand engagement. Therefore, TPP offers an appealing way to map the elusive *in vivo* substrate specificity of one of the largest paralogous family of proteins in bacteria (*E. coli* has 65–83 ABC transport systems), which are also ubiquitous in fungi, plants, and higher eukaryotes (Moussatova *et al*, 2008; Rees *et al*, 2009). Integrating stability and abundance protein levels in a single measurement for other stress conditions will offer further insight into cellular physiology and metabolic and genomic wiring.

The coherent melting behavior of the majority of protein complexes, as well as the destabilization of complex members upon *tolC* knockout, suggests that protein–protein interactions confer a mutual thermostabilizing effect in a cellular context [in line with previous *in vitro* (Despa *et al*, 2005) and theoretical work (Starzyk *et al*, 2016)]. Interestingly, many outliers could be explained by their subcellular localization—which likely reflects their structure [e.g., β-barrel structures being more stable (Burgess *et al*, 2008; Ureta *et al*, 2007)]—or by the formation of subcomplexes. These data might be helpful in structural analysis or to study the mechanisms of complex assembly. For example, recent structural data for the Bam complex (Bakelar *et al*, 2016; Gu *et al*, 2016; Han *et al*, 2016; Iadanza *et al*, 2016), which assembles β-barrels in the outer membrane and may also facilitate the translocation of lipoproteins to the surface (Cho *et al*, 2014; Konovalova *et al*, 2014), are in conflict with previous *in vivo* data on the topology of one of its components, BamC (Webb *et al*, 2012). Consistent with the less extensive association of BamC with the rest of the complex, the flexible orientation of its helix-grip domains in the crystal structure and their surface-exposed topology, we observe that BamC has clearly different melting behavior than the rest of the Bam subunits (being entirely thermostable).

All the above examples illustrate the unique view that TPP offers on cellular phenotypes when compared to other genome-wide approaches (Costanzo *et al*, 2016; Fuhrer *et al*, 2017; Mulleder *et al*, 2016; Nichols *et al*, 2011; Price *et al*, 2018). It seems tempting to suggest that TPP can be used in a similar manner to profile knock-out libraries in order to discover gene function (based on similarity

of phenotypic profiles). At this stage, even by profiling single knock-outs, TPP can provide immediate insights into the physical interaction of the partners of the missing protein and the direct cellular consequences caused by its absence. For example, we offer further evidence that YbhG is part of an efflux pump with TolC (Yamanaka *et al*, 2016; Babu *et al*, 2017) and that the absence of TolC leads to both accumulation of enterobactin and unfolded outer membrane β-barrel proteins in the periplasm. Moreover, the global view of the proteome allows explaining seemingly surprising effects, such as the increased resistance to aztreonam in the antibiotic hypersensitive *ΔtolC* strain, which we ascribe to the *micF*-dependent down-regulation of OmpF.

In addition to insights into cellular behavior, TPP also offers direct information into drug–target(s) interactions. Using TPP, we identified the primary targets of all three drugs tested and provided an estimate of the *in vivo* affinity toward their targets. Such facile target deconvolution has only recently been made possible with proteomics-based technologies, such as TPP (Savitski *et al*, 2014; Reinhard *et al*, 2015; Becher *et al*, 2016) or limited proteolysis (Leuenberger *et al*, 2017; Piazza *et al*, 2018). TPP has the advantage that it can also map the downstream consequences of target inhibition, in addition to direct targets (in lysate), since it can be performed in living cells. The results with the two bactericidal antibiotics we profiled as proof-of-principle point to a multifaceted cellular failure and stress alarm with hallmark effects on protein translation and quality control, in addition to specific effects of ampicillin (cell elongation/division) and of ciprofloxacin (DNA damage, histone-like proteins). We see no evidence for damaging reactive species being the main cause of stress after exposure to such bactericidal antibiotics (Kohanski *et al*, 2007; Ezraty *et al*, 2013; Keren *et al*, 2013; Liu & Imlay, 2013; Dwyer *et al*, 2014).

In conclusion, TPP in bacteria provides a powerful tool to understand protein–protein, protein–metabolite, and protein–drug interactions. Obtaining protein abundance and stability levels in a single measurement offers an informative global view of the cell, particularly when probed in distinct chemical, environmental, or genetic perturbations.

# Materials and Methods

### Bacterial strains and human cells

The BW25113 *E. coli* strain was used as the wild-type strain. The single deletion mutant strains used were from the Keio collection (Baba *et al*, 2006) after transduction into the wild-type strain using P1 transduction (Miller, 1972). *Escherichia coli* double-deletion mutants were obtained using P1 transduction (Miller, 1972), except *ΔsurAΔtolC,* which was obtained in M9 minimal medium supplemented with M9 salts, 0.4% glucose, 1 mM $MgSO_4$, 0.1 mM $CaCl_2$. HepG2 cells, used as a reference human cell, were bought from ATCC and tested negative for mycoplasma contamination.

### Thermal proteome profiling and sample preparation

The thermal proteome profiling protocol (Savitski *et al*, 2014; Franken *et al*, 2015; Reinhard *et al*, 2015; Becher *et al*, 2016) was modified to be compatible with *E. coli*.

Bacterial cells were grown overnight at 37°C in lysogeny broth (LB Lennox) and diluted 100-fold (except for experiments in exponential phase, which were diluted 2,000-fold) into 20 ml of fresh LB (except for experiments in exponential phase, in which 50 ml of medium was used, or in stationary phase, in which 10 ml of medium was used). Cultures were grown aerobically at 37°C with shaking until desired optical density at 578 nm ($OD_{578}$), generally ~0.5, except for exponential phase (~0.1) and transition to stationary phase (~2). For TPP experiments with drugs, the compound was added at five different concentrations to 20 ml of culture and incubated for 10 min; for all other experiments, the culture was used directly. Cells were pelleted at 4,000 × *g* for 5 min, washed with 10 ml PBS (containing the drug, for experiments with drugs), re-suspended in the same buffer to an $OD_{578}$ of 10, and 100 µl was aliquoted to ten wells of a PCR plate. The plate was centrifuged at 4,000 × *g* for 5 min, and 80 µl of the supernatant was removed prior to subjecting the plate to a temperature gradient for 3 min in a PCR machine (Agilent SureCycler 8800), followed by 3 min at room temperature. Cells were lysed with 30 µl lysis buffer [final concentration: 50 µg/ml lysozyme, 0.8% NP-40, 1× protease inhibitor (Roche), 250 U/ml benzonase, and 1 mM $MgCl_2$ in PBS] for 20 min, shaking at room temperature, followed by three freeze–thaw cycles (freezing in liquid nitrogen, followed by 1 min at 25°C in a PCR machine and vortexing). The plate was then centrifuged at 2,000 × *g* for 5 min to remove cell debris, and the supernatant was filtered at 500 × *g* for 5 min through a 0.45-µm 96-well filter plate (Millipore, ref: MSHVN4550) to remove protein aggregates. The flow-through was mixed 1:1 with 2× sample buffer (180 mM Tris pH 6.8, 4% SDS, 20% glycerol, 0.1 g bromophenol blue) and kept at −20°C until digested as described in "Protein digestion and peptide labeling".

For experiments with cell lysates, cells at $OD_{578}$ ~0.5 were washed and re-suspended to an $OD_{578}$ of 50 with lysis buffer (without NP-40), and lysed as described above. Drugs were then added to the lysate, and 20 µl was aliquoted to a PCR plate and subjected to the temperature gradient, as described above. NP-40 was then added to a final concentration of 0.8%, protein aggregates were removed, and samples were processed similar to whole-cell experiments.

To verify that NP-40 did not affect the solubilization of membrane proteins, cells at $OD_{578}$ ~0.5 were prepared and lysed as described above for experiments with living cells, except that no heat treatment was applied and SDS (instead of NP-40) was used at a final concentration of 2%.

HepG2 cells were grown in EMEM supplemented 10% fetal bovine serum at 37°C and a 5% $CO_2$ atmosphere. Cells were subcultured twice weekly with a 1:5 ratio. On the day of the experiment, cells were harvested and aliquoted to a PCR plate (500,000 cells/well). This plate was then processed as described above, except that the lysis buffer did not contain lysozyme, and lysis was performed for 1 h at 4°C.

### Protein digestion and peptide labeling

Proteins were digested according to a modified SP3 protocol (Hughes *et al*, 2014; Moggridge *et al*, 2018). Briefly, 2 µg of protein (for bacteria samples) or 10 µg of protein (for human samples) was diluted in water to 20 µl and added to the bead suspension [10 µg

of beads (Thermo Fischer Scientific—Sera-Mag Speed Beads, CAT# 4515-2105-050250, 6515-2105-050250) in 10 µl 15% formic acid and 30 µl ethanol]. Proteins were allowed to bind the beads for 15 min at room temperature with shaking and then washed four times with 70% ethanol. Next, proteins were digested overnight by adding 40 µl of digest solution (5 mM chloroacetamide, 1.25 mM TCEP, 200 ng trypsin, and 200 ng LysC in 100 mM HEPES pH 8). Peptides then were eluted from the beads and dried under vacuum. The peptides were then redissolved in 10 µl of water and labeled for 1 h at room temperature with 17 ng (bacteria) or 72 ng (humans) of TMT10plex (Thermo Fisher Scientific) dissolved in 4 µl of acetonitrile. The reaction was quenched with 4 µl of 5% hydroxylamine, and all conditions from one experiment were combined. Samples were desalted with solid-phase extraction by loading the samples onto a Waters OASIS HLB µElution Plate (30 µm), washing them twice with 100 µl of 0.05% formic acid, eluting them with 100 µl of 80% acetonitrile, and drying them under vacuum. Finally, samples were fractionated onto six (bacteria) or 12 (human) fractions on a reversed-phase C18 system running under high pH conditions (Hughes *et al*, 2014).

## Mass spectrometry-based proteomics

Samples were analyzed with liquid chromatography coupled to tandem mass spectrometry, as previously described (Becher *et al*, 2018). Briefly, peptides were separated using an UltiMate 3000 RSLCnano system (Thermo Fisher Scientific) equipped with a trapping cartridge (Precolumn; C18 PepMap 100, 5 µm, 300 µm i.d. × 5 mm, 100 Å) and an analytical column (Waters nanoEase HSS C18 T3, 75 µm × 25 cm, 1.8 µm, 100 Å). Solvent A was 0.1% formic acid in LC-MS grade water and solvent B was 0.1% formic acid in LC-MS grade acetonitrile. After loading the peptides onto the trapping cartridge (30 µl/min of solvent A for 3 min), elution was performed with a constant flow of 0.3 µl/min using a 60–120 min analysis time (with a 2–28% B elution, followed by an increase to 40% B, and re-equilibration to initial conditions). The LC system was directly coupled to a Q Exactive Plus mass spectrometer (Thermo Fisher Scientific) using a Nanospray-Flex ion source and a Pico-Tip Emitter 360 µm OD × 20 µm ID; 10 µm tip (New Objective). The mass spectrometer was operated in positive ion mode with a spray voltage of 2.3 kV and capillary temperature of 320°C. Full-scan MS spectra with a mass range of 375–1,200 m/z were acquired in profile mode using a resolution of 70,000 [maximum fill time of 250 ms or a maximum of 3e6 ions (automatic gain control, AGC)]. Fragmentation was triggered for the top 10 peaks with charge 2–4 on the MS scan (data-dependent acquisition) with a 30-s dynamic exclusion window (normalized collision energy was 32), and MS/MS spectra were acquired in profile mode with a resolution of 35,000 (maximum fill time of 120 ms or an AGC target of 2e5 ions).

## Drug sensitivity experiments

Cells were grown overnight in LB and diluted to an $OD_{578}$ of 0.04 with fresh 2× LB. Drug solutions were prepared in water from DMSO stocks and serially diluted twofold. An equal volume of cells and drug solution was then added to a 384-well plate. Cells were incubated at 37°C with shaking in a Tecan Safire2 plate reader, and $OD_{595}$ was monitored every 15 min. All growth curves were visually inspected for their quality, and the growth at 8 h was chosen to compare samples, as the time-point of the maximal growth for the wild-type non-stressed cells.

## Measurement of respiratory activity

Cells were grown overnight in LB and diluted 2,000-fold into 50 ml of fresh LB (for experiments in exponential phase) or 100-fold into 10 ml of fresh LB (for experiments in stationary phase). Cultures were grown aerobically at 37°C with shaking. Triphenyltetrazolium chloride was added to a final concentration of 0.02% when the cells reached an $OD_{578}$ ~0.01 (exponential phase) or ~0.5 (transition to stationary phase). The cells were then allowed to continue growing for 100 min to an $OD_{578}$ ~0.1 (exponential phase) or ~2 (transition to stationary phase). At that moment, the cells were pelleted at $4,000 \times g$ for 5 min, and triphenylformazan was extracted with isopropanol, adjusting the two cultures for cell number (by resuspending the cells to an $OD_{578}$ of 30). Cell debris was removed by centrifugation at $10,000 \times g$ for 10 min, and the absorbance of 200 µl of the supernatant was measured at 480 nm.

## Data analysis

### Protein identification and quantification
Mass spectrometry data were processed as previously described (Becher *et al*, 2018). Briefly, raw mass spectrometry files were processed with IsobarQuant (Franken *et al*, 2015), and the identification of peptide and protein was performed with Mascot 2.4 (Matrix Science) against the *E. coli* (strain K12) UniProt FASTA (Proteome ID: UP000000625), modified to include known contaminants and the reversed protein sequences (search parameters: trypsin; missed cleavages 3; peptide tolerance 10 ppm; MS/MS tolerance 0.02 Da; fixed modifications were carbamidomethyl on cysteines and TMT10plex on lysine; variable modifications included acetylation on protein N-terminus, oxidation of methionine, and TMT10plex on peptide N-termini).

### Thermal proteome profiling analysis
Thermal proteome profiling analysis was performed in R (R Core Team, 2017), similar to previously described (Franken *et al*, 2015). Briefly, all output data from IsobarQuant were normalized using variance stabilization (*vsn*; Huber *et al*, 2002). The *TPP* package (Franken *et al*, 2015) was then used for meltome (TPP-TR), growth phase (TPP-TR), target engagement of aztreonam (TPP-CCR), and 2D-TPP with drugs.

For TPP-TR experiments, melting curves were fitted to each protein, while for TPP-CCR and 2D-TPP experiments, dose–response curves were used. For growth phase experiments, a nonparametric approach to compare melting curves (available in the *TPP* package) was used, with $P < 0.001$ set as the threshold for significance. In the same experiment, abundance estimates were obtained from the top3 parameter (corresponding to the sum of the peak areas of the three most intense peptides for each protein), by considering the fraction of the signal belonging to the lowest temperature. A *limma* analysis (Ritchie *et al*, 2015) was used to assess significantly changed proteins, with $P < 0.05$ and a minimum twofold change as a threshold for significance. For the Δ*tolC* analysis, abundance and stability scores were calculated with a bootstrap

algorithm (Becher *et al*, 2018), together with a local FDR that describes the quality and the reproducibility of the score values (by taking into account the variance between replicates). A local FDR $P < 0.05$ and a minimum absolute score of 10 were set as thresholds for significance. To identify hits in 2D-TPP experiments with drugs, an FDR-controlled method for functional analysis of dose–response curves was used.

*Melting behavior of complexes analysis*

The average Euclidean distance between melting curves of proteins of the same complex was computed for complexes with at least two quantified protein members, as described in Tan *et al* (2018). Complexes were considered to melt coherently if their average Euclidean distance was below the $5^{th}$ percentile (FDR $P < 0.05$) of a distribution of average Euclidean distances from randomly generated complexes with the same number of proteins (drawn from the list of proteins in the meltome experiment).

## Data sources

Protein location in *E. coli* was collected from STEPdb v.2.0 (http://www.stepdb.eu/; Orfanoudaki & Economou, 2014). Protein complexes were collected from EcoCyc v.21.1 (https://ecocyc.org/; Keseler *et al*, 2017) or from Ori *et al* (2016). Gene ontologies were downloaded from http://geneontology.org/. Protein interactions were obtained from STRING database (https://string-db.org/; Szklarczyk *et al*, 2017) using Cytoscape (http://www.cytoscape.org/; Shannon *et al*, 2003).

## Data availability

The mass spectrometry proteomics data have been deposited to the ProteomeXchange Consortium via the PRIDE partner repository with the dataset identifier PXD009495.

**Expanded View** for this article is available online.

## Acknowledgements

We would like to thank George Kritikos for GO enrichment analysis and Pedro Beltrao for critically reading the manuscript and providing feedback. This work was supported by the European Molecular Biology Laboratory. AM was supported by a fellowship from the EMBL Interdisciplinary Postdoc (EI3POD) Programme under Marie Skłodowska-Curie Actions COFUND (grant number 664726).

## Author contributions

AM, JB, JH, and DH performed the experiments. AM, NK, FS, AT, and MMS performed the data analysis. AT, and MMS supervised the study. AM, AT, and MMS wrote the manuscript, with all authors critically reviewing it.

## Conflict of interest

The authors declare that they have no conflict of interest.

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
