## [Review Process File · Molecular Systems Biology]

Thermal proteome profiling in bacteria: probing protein state *in vivo*

André Mateus, Jacob Bobonis, Nils Kurzawa, Frank Stein, Dominic Helm, Johannes Hevler, Athanasios Typas and Mikhail M. Savitski.

Review timeline:

Submission date:	29 th January 2018
Editorial Decision:	16 th March 2018
Revision received:	9 th May 2018
Editorial Decision:	19 th June 2018
Revision received:	21 st June 2018
Accepted:	22 nd June 2018

Editor: Maria Polychronidou.

Transaction Report:

1st Editorial Decision

16th March 2018

Thank you again for submitting your work to Molecular Systems Biology. We have now heard back from the three referees who agreed to evaluate your study. As you will see below, the reviewers acknowledge that the datasets presented in this study are potentially relevant for the field. However, they raise a series of concerns, which we would ask you to address in a major revision of the manuscript.

Without repeating all the points listed below some of the more fundamental issues that need to be addressed are the following:

- Overall, the reviewers think that the part of the study describing the application of TPP to identify drug targets, drug mode of action and analyze bacterial resistance is potentially interesting. They point out however, that this part remains rather preliminary and that follow up providing further insights and better demonstrating that TPP can be used for the purposes described above would significantly enhance the impact of the study.
- The part describing the behavior of protein complexes in different subcellular compartments needs to be expanded, in order to provide further support for the related conclusions.
- The study needs to be better placed in the context of existing literature.
- The text should be edited in order to describe more clearly the novel contributions and main objectives of the study.
- In line with the comments of reviewer #3, we would ask you to make sure that the methodology is described in sufficient detail. All Materials and Methods should be described in the main text. Of course all other issues raised by the referees would need to be thoroughly addressed. As you

might already know, our editorial policy allows in principle a single round of major revision so it is essential to provide responses to the reviewers' comments that are as complete as possible.

REFeree REPORTS.

Reviewer #1:

Mateus et al. apply thermal proteome profiling to *E. coli*, both to describe the thermal resistance profile of this bacterial proteome, as well as to demonstrate the potential of TPP to identify metabolite (in this case antibiotic) binding proteins.

The study is well written, and I find no major flaws, neither in the conduction of the experiments, nor in their interpretation. In this sense I have no question that this study will make a valuable contribution to the scientific literature.

At the same time, the study also does not tell us many new things, with the exception that TPP works well on *E. coli* and that it is suitable on the application on two known antibiotics. Its largely descriptive: an increased or decreased stability of a protein in different conditions can have multiple reasons. Although the conclusions drawn are intuitive, literally none of the findings have been followed up with a second, independent technique, nor have the cases for differential protein stability been worked out in mechanistic detail. Its hence for the Editors to decide whether the manuscript meets the bar for MSB.

Minot point:

I'm a bit worried (only a bit) about one of the conclusions, the claim that *E. coli* proteome 'is' more heat-stable as the human proteome. This statement might be over-generalized, given the fact that the study has been conducted on one bacterial strain only. Here, adding a simple caveat might however do.

Reviewer #2:

The manuscript by Mateus et al. focuses on the thermal profiling of the *Escherichia coli* proteome under different conditions. The analysis relies on the well-established thermal proteome profiling (TPP) assay that measures thermal denaturation profiles of detectable proteins in a proteome by a quantitative mass spectrometry approach (PMID: 25278616). The study under review analyses the *in vivo* melting behaviour of *E. coli* proteins from different subcellular fractions obtained from a culture grown to exponential and stationary phase and upon drug treatment. The authors use the data to assess how thermal stability is influenced by cellular compartmentalization, protein complex formation and metabolic states. They also present the application of TPP to unravelling the effects of drug treatment.

Although the datasets generated in this study are potentially interesting, the study suffers from a general lack of focus and novelty and from some conceptual weaknesses, which severely detract from its potential impact. The main issue relates to the general purpose of the study. Based on the Introduction, it seems that the manuscript will focus on the use of TPP to characterize mechanisms of action of drugs and mechanisms of drug resistance. This is however not clearly addressed in the Results section, which describes a series of seemingly unrelated experiments. How these experiments are related to issues raised in the Introduction is not clear. First, some interesting observations from *E. coli* standard growth conditions are reported, most notably how the thermal stability of proteins depends on cellular location and how thermal melting profiles of certain complex subunits compare. Next, the authors analyse how abundances and thermal profiles of *E. coli* proteins change at stationary phase. The choice of this condition and how it relates to drug resistance and mechanism of action analysis are not clear. Last, the authors analyse proteome

thermal profiles upon drug treatment *in situ* and *in vivo* with the purpose of identifying drug targets and characterizing the effects of the drugs. These are potentially interesting observations, but it is not clear how the findings relate to each other or to the focus stated in the Introduction.

Another issue is novelty. A proteome-wide analysis of the thermal stabilities of *E. coli* proteins from cell lysates has already been conducted using another approach (PMID: 28232526). The insights generated by the *in vivo* application of TPP to *E. coli* are not readily apparent. The identification of subcomplexes from TPP data is rather anecdotal but potentially interesting. However, a recent study by the Nordlund group (PMID: 29439025) used the TPP approach to analyse the behaviour of complex subunits on a much larger scale and to predict complex membership using a more statistically sound approach. TPP has also already been used in the same study (PMID: 29439025) to characterize deregulated biological processes under different conditions based on altered protein stability data (PMID: 29439025). Drug target identification by TPP has already been reported in (more complex) human cell lines (PMIDs: 25278616, 27010513, 26711467). The idea of combining protein stability analyses *in situ* and *in vivo* and literature data to discriminate direct drug binding and downstream events is promising. However, *in vivo* and *in situ* TPP experiments have already been used as a proof of principle to reveal downstream effectors of Bcr-Abl signalling in leukaemia cells exposed to the drug dasatinib (PMIDs: 25278616). TPP in lysates has also been applied to explain some of the adverse effects of the drug Vemurafenib (PMIDs: 25278616). Based on this, the novelty of the presented data seems limited to the application of TPP to bacterial organisms. The idea of using TPP to probe bacterial resistance mechanisms is novel, but the presented data do not seem sufficient to support this claim since the identification of the proposed molecular mechanism mostly relies on protein expression data and previous literature evidence, rather than on TPP. Also, the lower apparent affinity of FtsI and MrcA for aztreonam observed using TPP in the *tolC* strain is not a direct measure of the intracellular concentration of aztreonam.

The analysis of protein abundances and protein stability in different growth phases and in a *tolC* knock-out strain is new and potentially interesting, but again the purpose of this analysis is not clearly explained. It is somewhat surprising that several proteins upregulated at stationary phase also appear to be thermally stabilized. Is this a real biological effect or could this be due to a bias of the TPP method (e.g., does it preferentially detect T_m variations for proteins that substantially change abundance)?

Now to the conceptual flaws: The authors imply that a different melting profile for subunits of protein complexes in a given compartment is indicative of the formation of sub-complexes. This claim is however not properly validated. Similarly, the authors imply in several sentences that the extent of thermal stabilization is a quantitative measure of protein activity (e.g., page 10, line 266; page 6, line 136; last sentence of the paper; page 12, line 357; page 1 line 310). This is based on the observation that a couple of dozens of proteins with increased thermostability at stationary phase compared to the exponential growth phase either have also increased abundance or are known to be needed at stationary phase. These data do not support the claim that thermostability reports on activity, and this claim is also likely to be completely wrong. For example, stability of enzymes might increase because they bind to molecules that block their active site, thus reducing their activity. Similarly, proteins with natively unfolded domains may require instability to exert their function/activity.

To summarize, the study does not have sufficient novelty, and it is not clear which overarching question it tries to address. Given also the conceptual flaws, I do not recommend it for publication.

Reviewer #3:

This is a very nice study establishing the recently developed thermal proteome profiling (TPP) method for studies of drug binding and cellular processes in *E. coli*. Previously the method has been applied for studies of mammalian proteomes. The most convincing demonstration of the stringency of the novel *E. coli* protocol is the study of several antibiotics in lysates and cells confirming interactions with known targets. In addition to drug interactions, the authors profile differences in growth states as well as effects of *TolC* downregulation, the later to simulate a scenario for resistance to antibiotics. Overall the data and analysis is convincing and in addition to the novel

protocol, there is quite significant novelty in the many proteins shown to be effected/shifted in the different processes. The manuscript should therefore potentially be well suited for publication in MSB. However, I feel that the conclusion around some of the results are not crisp and that the presentation could be improved.

The general melting behavior of the E coli proteome is discussed on page 4 but a previous study of the meltome of E coli by Leuenberger, using LiP-MS, is largely ignored - this study is only briefly referred to in the discussion. The authors should related their data and conclusions already in the result section (and in the Supplement) to this work including overall comparison of melting curves and on e.g. vulnerable proteins in different temperature ranges. It is important that novel methods are compare to prior methods, although in this case the different detection principles could be expected to give somewhat different results.

Average T_m and the impression from figure 1E do not appear to support major differences between proteins in cytoplasm, inner membrane and periplasm, the small statistical difference might be contributed by a few extreme proteins. Instead it is the outer membrane proteins that stands out. The "localization discussion" could therefore be deleted and instead focused on the highly stable outer membrane proteins. Here the average biphasic behavior of the OMPa in figure 1B is interesting but it is not clear how the biphasic melting behavior relate to the T_m distribution in Fig 1E and in particularly to the different subtypes of OMP (although briefly discussed in the text). The discussion could be improved and I would like to see a figure with melting curves of all (most) outer membrane where different subtypes (Beta-barrel proteins, lipoproteins etc) are differently colored to help the reader understand what is going on with different populations of OMPs.

The conclusion on complex co-precipitation being due to 1 or >1 location is also a bit forced. The physical basis for this is not well explained and as shown in Figure 2d, this could rather be that complexes in >1 locations are in the membrane, not the "multiple location conclusion". The membrane location could have significant effects on the behavior of membrane protein (vs non-membrane proteins) in the solubilisation and centrifugation steps of the TPP protocol. If this conclusion remains, the authors need to take the discussion on the physical basis for this phenomena much further. When many of the shifting proteins discussed in the manuscript are membrane proteins, they should also address methodological challenges of these proteins in TPP and also help the reader to distinguish these proteins in all the supplement .xls files (As in e.g. Supp Table 1), where there should be a column with cellular localization (as well as the full protein name).

Around the same time as this manuscript was submitted a paper came out in Science by Tan et al describing the co-precipitation of protein complexes and it's utility to study protein complexes in cells. This makes part of the discussion on protein complexes in the present work redundant and the authors should relate their analysis and conclusions to the Science paper and focus the presentation on what is novel and specific for the E coli protein complexes.

The quite dramatic difference in the TolC knockout experiment is interesting but potential artifacts need to be excluded. It appears to me the two E coli cell lines used in the study might not be in exactly the same genetic background? Also, a TolC knockout might rapidly accumulated compensating mutations changing expression levels and potentially also protein stability. The authors should address this problem and if there is any doubt that the genome sequences are (nearly) identical they should sequence the cells they used for the study to map differences.

When a key result of this work is the E coli TPP protocol, this protocol has to be presented in exquisite detail. In the current M&M it is only briefly presented and there are references to multiple previous papers which use different protocols. All details, including reagents, concentrations, time parameters, key instrumentation etc need to be given in the protocol to help the community to accurately adept the method.

Although TPP now provide an exciting new mean to identify proteins for which the biochemistry is changed in specific process, it remains challenging to establish conclusive structural mechanism for these shifts. In a number of places in the manuscript the authors relate shifts to known biochemical effects on the specific protein which is fine. However, for many of these proteins other mechanistic explanations are also possible. Ideally the authors should have provide addition data confirming the proposed mechanisms of some of the proteins they discuss but I realize the challenge of such follow

out studies, and in some cases no other methods than TPP/CETSA could give this information. Instead, when the authors are in the process of setting a standard for how to present TPP data, I suggest they carefully look through their statements on mechanisms for shifts and delete the most speculative ones, and also not shy away from discussing several possible mechanisms for a shift, when the mechanism will often remain uncertain until otherwise proven.

As far as I can see the PBP1-7 melting proteins discussed in the main text has other names in the supplementary tables. Might be other such cases so the authors should check that the naming is the same in manuscript and .xls files.

1st Revision - authors' response

9th May 2018

Authors Point by Point response.

- Overall, the reviewers think that the part of the study describing the application of TPP to identify drug targets, drug mode of action and analyze bacterial resistance is potentially interesting. They point out however, that this part remains rather preliminary and that follow up providing further insights and better demonstrating that TPP can be used for the purposes described above would significantly enhance the impact of the study.

We do understand the problem and this was largely our fault. In the first submission, we set the stage too much around drug discovery, but then results went into many different directions. We have now rewritten the Introduction and adapted the rest of the manuscript to reflect more the real aims of this paper, which is to present a flavor of the different ways TPP can be used to uncover new biology in bacteria. TPP can be used for MoA identification and understanding bacterial resistance (tolC is the major drug efflux pump in *E. coli*), but can be also used to probe protein activity, protein complexes and overall cellular physiology in different growth stages, genetic backgrounds or after chemical/environmental perturbations.

To highlight some of the experiments we did to strengthen the manuscript and to reinforce the point that TPP can be used to capture new biology:

- we validated the performance of the TPP method itself for capturing melting curves of membrane proteins (comparison of NP40 and SDS extraction; latter is new).
- we provide orthogonal experimental confirmation that the increase in thermostability of members of the respiratory complex in the transition to stationary phase translates into higher respiratory activity.
- we provide orthogonal experimental evidence that increased SurA thermostability (and overall periplasmic chaperones) in $\Delta tolC$ is due to increased activity; this is corroborated by the essentiality of a periplasmic chaperone (SurA) and the activation of envelope surveillance systems in this condition.
- we performed TPP in lysate in order to make a detailed comparison to LiP-MS. In this context we discover that the ribosome is uniquely and profoundly modulated in its thermal stability by magnesium concentrations as compared to the rest of the entire proteome.

We also put and describe our novel resistance mechanism findings in a better context – that is that Aztreonam uses OmpF to enter the cell. This finding is novel and validated by additional experiments. We feel that we did not do it justice in the text in the first submission.

- The part describing the behavior of protein complexes in different subcellular compartments needs to be expanded, in order to provide further support for the related conclusions.

We now provide clear cut statistical evidence for different stabilities in different compartments. We also convincingly show that our methodology extracts all membrane proteins, as the mild detergent NP-40 extracts equally well membrane proteins as the strong detergent SDS. This solidifies the point that our assessment of thermal stability of membrane and other proteins is complete and unbiased.

- The study needs to be better placed in the context of existing literature.

In the revised manuscript, we perform a detailed comparison to the complementary LiP-MS methodology as requested by reviewer 3. The results from the two methods largely agree. Since LiP-MS is restricted to lysate we had to perform additional TPP lysate experiments to do the comparison. On the other hand, the nature of TPP, which allows for its use in living cells, is the premise for our ability to probe cellular physiology and protein state upon chemical, genetic or environmental perturbations.

We also now cite and use algorithms from the Tan et al paper. In response to the reviewers, we make it very clear why our complex analysis in *E. coli* adds novelty to the TPP complex discussion presented in this study. Thus, we do not feel that the Tan study takes away anything from our manuscript. Nevertheless, we would still like to point out that the Tan et al study came out after our study was submitted.

- The text should be edited in order to describe more clearly the novel contributions and main objectives of the study.

We made significant changes to the text (introduction and results) to address these points. For details, please see response to reviewers.

- In line with the comments of reviewer #3, we would ask you to make sure that the methodology is described in sufficient detail. All Materials and Methods should be described in the main text.

We now provide a detailed description as instructed.

REFEREE REPORTS.

Reviewer #1:

Mateus et al. apply thermal proteome profiling to *E. coli*, both to describe the thermal resistance profile of this bacterial proteome, as well as to demonstrate the

potential of TPP to identify metabolite (in this case antibiotic) binding proteins.

The study is well written, and I find no major flaws, neither in the conduction of the experiments, nor in their interpretation. In this sense I have no question that this study will make a valuable contribution to the scientific literature.

At the same time, the study also does not tell us many new things, with the exception that TPP works well on *E. coli* and that it is suitable on the application on two known antibiotics. Its largely descriptive: an increased or decreased stability of a protein in different conditions can have multiple reasons. Although the conclusions drawn are intuitive, literally none of the findings have been followed up with a second, independent technique, nor have the cases for differential protein stability been worked out in mechanistic detail. Its hence for the Editors to decide whether the manuscript meets the bar for MSB.

We thank the reviewer for acknowledging the technical quality of our work. We believe that the expansion of the TPP methodology in bacteria is on its own of great importance, as it allows for probing for Mechanism-of-Action in an era of antibiotic crisis, especially for Gram-negative bacteria. In addition, we believe that we go further than this and show that TPP can be used more generally to study bacterial physiology, by:

- delivering melting curves for each protein of the proteome (as shown by our meltome analysis), which offer insight into protein structure and its dependence on cellular location and protein complexes
- providing information about the metabolic state of the cell (as shown by the experiments in exponential and stationary phase – which put TPP at the interface between proteomics and metabolomics in a single measurement)
- offering a new way to study protein complexes (as shown by our results that protein complexes co-melt, as well as the destabilization of complex members upon the deletion of a co-member, which, to the best of our knowledge, is performed for the first time).

We understand that this was not clearly communicated in the manuscript and have rewritten it (particularly, the introduction) to make the point that TPP goes beyond MoA identification.

We also would like to point out that our interpretations about changes in protein thermostability in different conditions are mostly based on recapitulating established literature (e.g. TnaA growth phase behavior, (sub)-complexes, enterobactin toxicity in *tolC* mutant, TolC efflux pump complexes, etc), rather than intuition. In terms of follow-up, in the first submission, we had shown that OmpF transports aztreonam. This was to our knowledge, demonstrated for the first time and required follow-up experiments. We have now performed additional follow-ups to confirm that stabilization of subunits of respiratory complexes in the transition to stationary phase is because of higher respiratory activity, and that stabilization of periplasmic chaperones in $\Delta tolC$ is due to higher presence of cargo (= periplasmic stress) and thus activity of chaperones.

Action taken: rewrote the manuscript (particularly, introduction) to highlight the novelty that comes from TPP. We provide 2 additional follow-up experiments (increased respiration in transition to stationary phase, tolC-surA synthetic lethality) in addition to the already existing example on Aztreonam using OmpF to enter the cells.

Minor point:

I'm a bit worried (only a bit) about one of the conclusions, the claim that *E. coli* proteome 'is' more heat-stable as the human proteome. This statement might be over-generalized, given the fact that the study has been conducted on one bacterial strain only. Here, adding a simple caveat might however do.

The reviewer is right, we have now made the distinction in the results section that this applies to the strain, BW25113.

Action taken: clarified which strain was used in the results section (p.4, line 91).

Reviewer #2:

The manuscript by Mateus et al. focuses on the thermal profiling of the *Escherichia coli* proteome under different conditions. The analysis relies on the well-established thermal proteome profiling (TPP) assay that measures thermal denaturation profiles of detectable proteins in a proteome by a quantitative mass spectrometry approach (PMID: 25278616). The study under review analyses the *in vivo* melting behaviour of *E. coli* proteins from different subcellular fractions obtained from a culture grown to exponential and stationary phase and upon drug treatment. The authors use the data to assess how thermal stability is influenced by cellular compartmentalization, protein complex formation and metabolic states. They also present the application of TPP to unravelling the effects of drug treatment.

Although the datasets generated in this study are potentially interesting, the study suffers from a general lack of focus and novelty and from some conceptual weaknesses, which severely detract from its potential impact. The main issue relates to the general purpose of the study. Based on the Introduction, it seems that the manuscript will focus on the use of TPP to characterize mechanisms of action of drugs and mechanisms of drug resistance. This is however not clearly addressed in the Results section, which describes a series of seemingly unrelated experiments. How these experiments are related to issues raised in the Introduction is not clear. First, some interesting observations from *E. coli* standard growth conditions are reported, most notably how the thermal stability of proteins depends on cellular location and how thermal melting profiles of certain complex subunits compare. Next, the authors analyse how abundances and thermal profiles of *E. coli* proteins change at stationary phase. The choice of this condition and how it relates to drug resistance and mechanism of action analysis are not clear. Last, the authors analyse proteome thermal profiles upon drug treatment *in situ* and *in vivo* with the purpose of identifying drug targets and characterizing the effects of the drugs. These are

potentially interesting observations, but it is not clear how the findings relate to each other or to the focus stated in the Introduction.

We thank the reviewer for pointing out the inconsistency of our drug-centered introduction and the results that verge towards different directions. This was indeed our mistake. As highlighted in our response to reviewer #1, we believe that TPP offers much more than an approach to identify drug mode of action, and that is what we try to illustrate with the series of experiments we present. Experiments are carefully chosen to give a flavor of the different aspects of protein biology, TPP can shed light into – i.e. protein complexes and protein activity at different growth phases or genetic, chemical or environmental perturbations. We hope that the rewritten introduction better puts this study in context.

Action taken: rewrote the introduction to correctly frame the scope of this study (p. 3).

Another issue is novelty. A proteome-wide analysis of the thermal stabilities of *E. coli* proteins from cell lysates has already been conducted using another approach (PMID: 28232526). The insights generated by the *in vivo* application of TPP to *E. coli* are not readily apparent.

We think that LiP-MS and TPP are complementary (as reviewer #3 seems to agree). Limited proteolysis provides peptide-level resolution on protein unfolding (this requires extensive analysis to obtain sufficient sequence coverage), but can only be applied to lysates. Conversely, TPP is a simpler approach to study cellular physiology as it can be applied *in vivo* (offering information on the effects of chemical, genetic and environmental perturbations not only on protein thermal stability, but also abundance in a single measurement). LiP-MS can thus provide unique insight on what part of the protein sequence drives stability changes of a protein in lysates. TPP, as we hope we demonstrate in this manuscript, can provide unique insights into cellular behaviors which require experiments to be performed *in vivo* – e.g. drug resistance mechanisms (Aztreonam using OmpF to enter cell), metabolic activity (increased respiration during transition to stationary phase), active protein quality control (SurA being stabilized in the *tolC* mutant as an indication of increased activity due to periplasmic stress – corroborated by its essentiality in these conditions), change in protein state upon growth phase (e.g. TnaA being freed from polar-located inactive foci during transition to stationary phase), changes in cellular physiology upon perturbation (accumulation of toxic enterobactin in periplasm in *tolC* mutants), etc. Thus, in our view both approaches have unique strengths and we foresee that combinations of both will have great impact on pathogen biology.

We have also now performed a comparison with the study by Leuenberger et al., which included two additional meltome experiments in lysate.

Action taken: included two additional meltome experiments in lysate (at 1 mM or 10 mM MgCl₂) to compare our results with the study by Leuenberger et al. We also include 2 new follow-ups (in total 3 follow-ups, with existing info) on indications

areas where the biological insight came directly from the *in vivo* application of TPP.

The identification of subcomplexes from TPP data is rather anecdotal but potentially interesting. However, a recent study by the Nordlund group (PMID: 29439025) used the TPP approach to analyse the behaviour of complex subunits on a much larger scale and to predict complex membership using a more statistically sound approach. TPP has also already been used in the same study (PMID: 29439025) to characterize deregulated biological processes under different conditions based on altered protein stability data (PMID: 29439025).

The study by Tan et al. appeared in the literature after our initial submission and we were happy to see that our observations could be confirmed by another group. Following the reviewer's suggestion, we have now redone our analysis using the same approach as in Tan et al. (i.e., Euclidean distances between melting curves). Using this approach, we have also been able to calculate the distances between all protein pairs of the same complex, which allows for a better visualization of subcomplexes (we now include this as Figure EV2).

We believe that our analysis goes beyond the analysis performed by Tan et al., since:

1. it is performed in *E. coli* for the first time
2. it includes membrane proteins (not present in Tan et al.), which allowed us to observe that complexes across subcellular compartments do not co-melt. Only 4% of the complexes that are located across compartments significantly co-melt, compared to 38% of the complexes that are in the same compartment. Note that the latter number is very similar to what observed by Tan et al. for a completely different organism (37%)
3. we perform an analysis of distinctly behaving subcomplexes and validate it with previous literature knowledge
4. we establish that the absence of one member of a protein complex affects the thermostability of the rest of its members ($\Delta tolC$ mutant – decreased thermostability of all different interacting partners)

Action taken: reanalyzed data on protein complexes according to the approach by Tan et al. (p. 7) and included a new analysis on subcomplexes (Figure EV2).

Drug target identification by TPP has already been reported in (more complex) human cell lines (PMIDs: 25278616, 27010513, 26711467). The idea of combining protein stability analyses *in situ* and *in vivo* and literature data to discriminate direct drug binding and downstream events is promising. However, *in vivo* and *in situ* TPP experiments have already been used as a proof of principle to reveal downstream effectors of Bcr-Abl signalling in leukaemia cells exposed to the drug dasatinib (PMIDs: 25278616). TPP in lysates has also been applied to explain some of the adverse effects of the drug Vemurafenib (PMIDs: 25278616). Based on this, the novelty of the presented data seems limited to the application of TPP to bacterial organisms.

We completely agree with the author that target identification by TPP has been previously performed in human cells, but we find that the expansion to bacteria warrants some merit for both technical (lysis conditions; different time-scale of responses) and biological reasons (human cells cannot be used to identify antibacterial compounds; methodology can be moved to genetically non-tractable bacteria). As we think that TPP can be a very powerful tool for probing MoA of antibacterials in the future, we opted to provide two examples, which would validate the current protocol and show that the modifications introduced do not impact a fundamental assumption of this approach (i.e., that targets are stabilized upon drug binding).

The idea of using TPP to probe bacterial resistance mechanisms is novel, but the presented data do not seem sufficient to support this claim since the identification of the proposed molecular mechanism mostly relies on protein expression data and previous literature evidence, rather than on TPP.

We agree with the reviewer and have removed claims that suggest that TPP can be used at this point to probe resistance without previous literature knowledge. We still believe that using TPP in conjunction with previous knowledge, we can understand mechanisms of resistance better, as illustrated by our observation of the shut-down of the OmpF (porin) expression upon *tolC* knock-out, which led us to hypothesize and prove that Aztreonam is using OmpF to enter the cells and this is the reason that *tolC* mutants become resistant to Aztreonam. Similarly, TPP could detect the stabilization of a beta-lactamase (AmpC) upon ampicillin treatment, which implies that in the future TPP could be used to identify novel deactivating enzymes of antibiotics in the cell.

Action taken: removed major claims that TPP can be used to probe resistance on its own.

Also, the lower apparent affinity of FtsI and MrcA for aztreonam observed using TPP in the *tolC* strain is not a direct measure of the intracellular concentration of aztreonam.

We agree that this measure is indirect, but we see no other explanation for this shift. We have rephrased this statement to only suggest that it could be caused by lower intracellular concentration.

Action taken: adjusted our statement to this “most likely” being the mechanism (p.10, line 289)

The analysis of protein abundances and protein stability in different growth phases and in a *tolC* knock-out strain is new and potentially interesting, but again the purpose of this analysis is not clearly explained.

Both serve a different purpose, which is now explained better in text. We measure protein thermostability and abundance in different growth phases to illustrate that we can capture known changes in protein levels and state during this well

characterized transition. As with growth phase transitions, TPP can be used similarly to probe less well-mapped environmental changes (e.g. media shifts), cell-cycle or developmental stages of bacterial growth (e.g. biofilm).

We perform the tolC experiment as a validation to our observation that protein complexes co-melt (i.e., do proteins stabilize other members of a complex? Or do they just co-melt by chance?). We chose TolC, as a widely studied member of multiple complexes involved in drug resistance. Even though it is widely studied, we provide further evidence to a potential new complex including this protein (YbhGFSR), since it behaves in a similar manner to other complex members and has been pulled-down with TolC (Babu, Nat Biotech, 2017). In addition, tolC serves as an example of how TPP can be used to probe cellular changes in mutants – assessing the effects of single-gene knockouts in the proteome state (abundance and stability).

Action taken: clarified in the text why these experiments were performed (p. 6 and 8).

It is somewhat surprising that several proteins upregulated at stationary phase also appear to be thermally stabilized. Is this a real biological effect or could this be due to a bias of the TPP method (e.g., does it preferentially detect T_m variations for proteins that substantially change abundance)?

We believe that this is a real biological effect, which is caused by the cell needing those enzymes and transporters in stationary phase. Therefore, cells upregulate their expression and there is increased flux through them, so they get stabilized. There are though exceptions to this behavior: TnaA gets upregulated but stability is impaired, consistent with literature evidence and a biophysical transition of the protein during transition to stationary phase (protein is released from inactive foci and becomes active during this stage).

Please also note that in the $\Delta tolC$ experiment, the opposite general relation exists (i.e. downregulation leads to stabilization; see figure below), confirming that this relationship is not a TPP bias and that each stress induces a particular response.

Now to the conceptual flaws: The authors imply that a different melting profile for subunits of protein complexes in a given compartment is indicative of the formation of sub-complexes. This claim is however not properly validated.

We have now included an additional supplementary figure (Figure EV2), in which we determine the Euclidean distances of all protein pairs of the same complex (a lower value indicates that the members co-melt). The examples provided in Figure 2 are not just based on our observations, but confirm existing literature knowledge on the existence of such subcomplexes. We try to properly reference all of these studies and briefly describe their findings. Thus, we find that these provide clear validation that TPP can detect the existence of such subcomplexes. We have now tried to make this point clearer in the text:

Action taken: clarified in the text that the examples given are based on prior literature knowledge (p. 8, line 219) and new supplementary figure (Figure EV2).

Similarly, the authors imply in several sentences that the extent of thermal stabilization is a quantitative measure of protein activity (e.g., page 10, line 266; page 6, line 136; last sentence of the paper; page 12, line 357; page 1 line 310). This is based on the observation that a couple of dozens of proteins with increased thermostability at stationary phase compared to the exponential growth phase either have also increased abundance or are known to be needed at stationary phase. These data do not support the claim that thermostability reports on activity, and this claim is also likely to be completely wrong. For example, stability of enzymes might increase because they bind to molecules that block their active site, thus reducing their activity. Similarly, proteins with natively unfolded domains may require instability to exert their function/activity.

We thank the reviewer for pointing this out. Indeed, changes in protein activity could result to either decrease or increase (or even in no detectable change) in its thermostability. We have now removed general claims that an increase in stability always corresponds to an increase in activity. Nevertheless, in many cases, increased stability recapitulates literature knowledge about increased activity of same protein/protein complex in same conditions. We still refer to such examples, providing the adequate citation. In addition, we now include two additional experiments, in which we show that the stabilization of members of the respiratory complex in the transition to stationary phase does in fact relate to increased respiratory activity, and that the stabilization of periplasmic chaperones in $\Delta tolC$ is due to their activity (synthetic lethality of $\Delta surA$ and $\Delta tolC$).

Action taken: performed additional experiments showing higher respiratory activity in the transition to stationary phase, essentiality of *surA* in $\Delta tolC$, and removed general claims that changes in thermal stability generally correspond to changes in protein activity.

To summarize, the study does not have sufficient novelty, and it is not clear which overarching question it tries to address. Given also the conceptual flaws, I do not recommend it for publication.

We thank the reviewer for the comments, which have increased the quality of our manuscript after the extensive changes and the additional experiments. We hope that the additional evidence and explanations we provide clarify concerns about conceptual flaws and illustrate the novelty of this study. We believe that the direct use of the datasets provided in this manuscript, as well as potential new

applications of TPP in bacterial cells, should be of interest for the broad audience of *Molecular Systems Biology*.

Reviewer #3:

This is a very nice study establishing the recently developed thermal proteome profiling (TPP) method for studies of drug binding and cellular processes in *E. coli*. Previously the method has been applied for studies of mammalian proteomes. The most convincing demonstration of the stringency of the novel *E. coli* protocol is the study of several antibiotics in lysates and cells confirming interactions with known targets. In addition to drug interactions, the authors profile differences in growth states as well as effects of TolC downregulation, the later to simulate a scenario for resistance to antibiotics. Overall the data and analysis is convincing and in addition to the novel protocol, there is quite significant novelty in the many proteins shown to be effected/shifted in the different processes. The manuscript should therefore potentially be well suited for publication in MSB. However, I feel that the conclusion around some of the results are not crisp and that the presentation could be improved.

We value the enthusiasm, and thank the reviewer for the insightful comments.

The general melting behavior of the *E. coli* proteome is discussed on page 4 but a previous study of the meltome of *E. coli* by Leuenberger, using LiP-MS, is largely ignored - this study is only briefly referred to in the discussion. The authors should related their data and conclusions already in the result section (and in the Supplement) to this work including overall comparison of melting curves and on e.g. vulnerable proteins in different temperature ranges. It is important that novel methods are compare to prior methods, although in this case the different detection principles could be expected to give somewhat different results.

We apologize for this omission, which was due to the fact that we had not determined the *E. coli* meltome in a lysate (to be able to directly compare with the results from LiP-MS). We have now done this and performed the analysis suggested (p. 5, line 132; Figure 1F-G). In summary, we generally observed a good correlation between the two methodologies, with the exception of ribosomal proteins (which showed higher stability when analyzed by LiP). However, we noticed that the buffers used by Leuenberger et al. contained higher amounts of magnesium, which is known to increase the thermal stability of ribosomal proteins. This prompted us to perform yet another lysate meltome determination using similar levels of magnesium to those used by Leuenberger et al. The higher concentration of magnesium had a very specific effect on ribosome stability as compared to the rest of the proteome which led to a much improved correlation with the LiP methodology.

Action taken: included two additional meltome experiments in lysate (at 1 mM or 10 mM MgCl₂) to compare our results with the study by Leuenberger et al (Figure 1F-G).

Average T_m and the impression from figure 1E do not appear to support major

differences between proteins in cytoplasm, inner membrane and periplasm, the small statistical difference might be contributed by a few extreme proteins. Instead it is the outer membrane proteins that stands out. The "localization discussion" could therefore be deleted and instead focused on the highly stable outer membrane proteins.

We respectfully disagree with this comment, as this is indeed a statistically significant difference, and it is not only due to some outliers but the mean, median and IQR of distribution is considerably different between proteins in different compartments (e.g. cytoplasmic compared to any other 3 groups). However, we thank the reviewer for pointing out that this important point should be made clearer. Therefore, we now include the results of the statistical test in the manuscript (p. 5, line 120). We also now also provide an alternative representation of the data in the supplement (Figure EV1H), in which we hope that the very clear shift in the distribution of melting temperatures also for proteins in the inner membrane and periplasm becomes even more evident. We have therefore kept the overall discussion unchanged, but now refer to statistical significances and supplementary figure in the main manuscript.

Action taken: performed statistically analysis to show that melting points of proteins in inner membrane and periplasm proteins are different from the remaining compartments (p. 5, line 120), and included a new supplementary figure as an alternative representation of the data (where differences become more apparent; Figure EV1H).

Here the average biphasic behavior of the OMPa in figure 1B is interesting but it is not clear how the biphasic melting behavior relate to the T_m distribution in Fig 1E and in particularly to the different subtypes of OMP (although briefly discussed in the text). The discussion could be improved and I would like to see a figure with melting curves of all (most) outer membrane where different subtypes (Beta-barrel proteins, lipoproteins etc) are differently colored to help the reader understand what is going on with different populations of OMPs.

We have now included the proteins that melt at a temperature higher than the highest temperature tested (non-melting) in Fig. 1E. In addition, we have included a supplementary figure with the fraction of non-melting proteins in each compartment (Figure EV1I). Finally, as suggested by the reviewer, we have also included a figure in which the melting curves of integral proteins or lipoproteins are differently colored (Figure EV1G). As already written in the manuscript, there is a tendency for integral membrane proteins (beta-barrel proteins) to be more thermostable than lipoproteins ($p=0.031$, Mann-Whitney test). However, since this is not such a strong difference (some lipoproteins are as/more stable than integral membrane proteins), we refrained from speculating further about which structural features confer thermal stability. As a note, according to recent literature (e.g. PMIDs 25267629, 25525882), lipoproteins may traverse the pores of beta-barrel proteins to access the cell surface; this bound state with integral OM proteins, may make them more thermostable.

Action taken: included two new supplementary figures (Figure EV1G and I) to show the behavior of outer membrane proteins.

The conclusion on complex co-precipitation being due to 1 or >1 location is also a bit forced. The physical basis for this is not well explained and as shown in Figure 2d, this could rather be that complexes in >1 locations are in the membrane, not the "multiple location conclusion". The membrane location could have significant effects on the behavior of membrane protein (vs non-membrane proteins) in the solubilisation and centrifugation steps of the TPP protocol.

First, to verify that the mild detergent used (NP40) did not introduce a bias when solubilizing the membranes, we performed an experiment in which we lysed cells with NP40 or with a strong detergent (SDS) and ran these samples in a single TMT experiment. We observed that the levels of extracted proteins were very similar between the two detergents (Figure EV1C).

Then, we reanalyzed the data on protein complexes in light of the new approach proposed by Tan et al. Since we can now calculate which protein complexes significantly co-melt, it becomes clearer that there is a strong depletion in co-melting complexes that contain proteins in more than one location (Figure 3B). To ensure that this is not just driven by the presence of membrane proteins in the complexes with >1 location, we have further analyzed the Euclidean distances of these complexes, separating the ones that include membrane proteins to the ones that do not. From the figure below (all represented complexes are in more than one location), it is possible to see a non-uniform behavior of complexes that include membrane proteins (some melt tightly and others do not), and that the few complexes without membrane proteins do not melt coherently. A complex with >1 location that does not contain membrane proteins in our data refers to complexes that contain cytoplasmic, inner membrane and periplasmic proteins in reality, but the inner membrane members are not detected by MS (thus also the periplasmic and cytoplasmic members do not co-melt). Taken together, this reinforces our conclusions that it is the multiple location that leads to less coherent melting behavior. This is consistent with periplasmic proteins being overall more thermostable than cytoplasmic ones (maybe due to the reducing environment in periplasm and the extensive stabilization of disulfide bonds), and with outer membrane proteins being more thermostable than inner membrane ones. Thus location-dependence is not a mere difference between stability of soluble and membrane proteins.

Action taken: performed an additional experiment comparing protein solubility upon NP40 or SDS extraction to show that the presented protocol (including NP40) does not impair the solubilization of membrane proteins.

If this conclusion remains, the authors need to take the discussion on the physical basis for this phenomena much further.

At this point, we feel that we can only offer hypotheses to explain this difference. It seems that stabilization induced by protein location (which is encoded in the protein sequence) overrides the effect of stabilization by other complex members. It could be that different diffusion dynamics in membranes and in soluble compartments define sub-complex formation; with complexes spanning different compartments to rarely form stable complexes that move together as units (there is some literature evidence on this). It also seems that protein stability has an overall location dependency that is not binary (membrane vs. soluble): this can be due to the different lipid composition and fluidity of the two membranes, the different structural requirements of proteins in the two membranes (e.g. all integral outer membrane proteins and beta-barrels), the reducing environment in periplasm (which is supported by disulfide bond catalyzing enzymes present in this compartment).

At this stage, we opted to leave this discussion out of the main text as too hypothetical, but if the reviewer and editor think is important to include, we can include it.

When many of the shifting proteins discussed in the manuscript are membrane proteins, they should also address methodological challenges of these proteins in TPP and also help the reader to distinguish these proteins in all the supplement .xls files (As in e.g. Supp Table 1), where there should be a column with cellular localization (as well as the full protein name).

As detailed above, the use of a mild detergent does not seem to bias the extraction/quantification of proteins. Further, we can observe stability changes in

membrane proteins when changing the genetic background (e.g., the destabilization of AcrAB (membrane proteins) in the $\Delta tolC$) Further support to the fact that TPP can detect stability changes in membrane comes from the original publication using NP40 and TPP (Reinhard et al., Nat Methods, 2015). Therefore, we believe that the current setup does not include any methodological challenges related to membrane proteins.

Action taken: included a new experiment comparing SDS and NP40 solubilized proteins, indicating that this does not impact membrane protein extraction, and added the additional columns suggested by the reviewer.

Around the same time as this manuscript was submitted a paper came out in Science by Tan et al describing the co-precipitation of protein complexes and it's utility to study protein complexes in cells. This makes part of the discussion on protein complexes in the present work redundant and the authors should relate their analysis and conclusions to the Science paper and focus the presentation on what is novel and specific for the E coli protein complexes.

We transcribe here our response to reviewer 2, explaining what we have changed in this analysis and what we consider novel in this manuscript compared to Tan et al. (which came out after our work was submitted).

“The study by Tan et al. appeared in the literature after our initial submission and we were happy to see that our observations could be confirmed by another group. Following the reviewer’s suggestion, we have now redone our analysis using the same approach as in Tan et al. (i.e., Euclidean distances between melting curves). Using this approach, we have also been able to calculate the distances between all protein pairs of the same complex, which allows for a better visualization of subcomplexes (we now include this as Figure EV2).

We believe that our analysis goes beyond the analysis performed by Tan et al., since:

1. it is performed in *E. coli* for the first time
2. it includes membrane proteins (not present in Tan et al.), which allowed us to observe that complexes across subcellular compartments do not co-melt. Only 4% of the complexes that are located across compartments significantly co-melt, compared to 38% of the complexes that are in the same compartment. Note that the latter number is very similar to what observed by Tan et al. for a completely different organism (37%)
3. we perform an analysis of distinctly behaving subcomplexes and validate it with previous literature knowledge
4. we establish that the absence of one member of a protein complex affects the thermostability of the rest of its members ($\Delta tolC$ mutant – decreased thermostability of all different interacting partners)

Action taken: reanalyzed data on protein complexes according to the approach by Tan et al. (p. 7) and included a new analysis on subcomplexes (Figure EV2).”

The quite dramatic difference in the TolC knockout experiment is interesting but potential artifacts need to be excluded. It appears to me the two E coli cell lines used in the study might not be in exactly the same genetic background? Also, a TolC knockout might rapidly accumulated compensating mutations changing expression levels and potentially also protein stability. The authors should address this problem and if there is any doubt that the genome sequences are (nearly) identical they should sequence the cells they used for the study to map differences.

We apologize for the initial omission in the manuscript, but to ensure a similar background we have P1-retransduced the *tolC* knockout into the wildtype strain at the beginning of the study. This ensures that genetic backgrounds between 2 strains are in principle isogenic (besides $\Delta tolC$ mutation). We make now this clear in the methods section.

We also recapitulate many known phenotypes of the *tolC* mutant – including enterobactin accumulation in periplasm, envelope stress responses, higher motility, lower OmpF levels, which all confirm that we are working with a mutant that has not accumulated compensating mutations.

Action taken: clarified in the methods section how strains was constructed (p. 15, line 441).

When a key result of this work is the E coli TPP protocol, this protocol has to be presented in exquisite detail. In the current M&M it is only briefly presented and there are references to multiple previous papers which use different protocols. All details, including reagents, concentrations, time parameters, key instrumentation etc need to be given in the protocol to help the community to accurately adept the method.

The reviewer is right. Since the TPP methodology has been extensively described in the past (including a protocol paper), we initially opted to highlight only the key differences that we introduced when adapting the protocol to *E. coli*. However, in light of this comment, we have now extensively expanded the methods section to include all the details on how we performed the experiments.

Action taken: expanded the methods section (p.15, line 450).

Although TPP now provide an exciting new mean to identify proteins for which the biochemistry is changed in specific process, it remains challenging to establish conclusive structural mechanism for these shifts. In a number of places in the manuscript the authors relate shifts to known biochemical effects on the specific protein which is fine. However, for many of these proteins other mechanistic explanations are also possible. Ideally the authors should have provide addition data confirming the proposed mechanisms of some of the proteins they discuss but I realize the challenge of such follow out studies, and in some cases no other methods than TPP/CETSA could give this information. Instead, when the authors are in the process of setting a standard for how to present TPP data, I suggest they careful look through their statements on mechanisms for shifts and delete the most

speculative ones, and also not shy away from discussing several possible mechanism for a shift, when the mechanism will often remain uncertain until otherwise proven.

As highlighted in response to reviewer 2, we have removed general claims for some of the shifts observed (e.g., that stabilization is generally caused by increased activity). To show however that this is often a reason for stabilization of proteins, besides highlighting cases where our data recapitulate established knowledge (i.e. changes in protein stability match previous reports for change in protein activity under same conditions), we have performed an additional experiment to demonstrate higher respiratory activity in stationary phase (Figure 2C) – which correlates with increased stability of the respiratory complex proteins in this growth phase. We also now show that the stabilization of periplasmic chaperones in a *tolC* mutant are indicative of their increased action based on the a) activation of multiple envelope stress responses (which in several cases detected unfolded proteins in periplasm) and b) the essentiality of the chaperone SurA in this background.

We would also like to point out that the follow-up experiments performed by aztreonam were based on the TPP data and led us to discover that OmpF is required for the import of this drug.

Action taken: We performed additional experiments showing higher respiratory activity in the transition to stationary phase and increased importance of SurA activity in $\Delta tolC$ cells. We also removed general claims that changes in thermal stability generally correspond to changes in protein activity.

As far as I can see the PBP1-7 melting proteins discussed in the main text has other names in the supplementary tables. Might be other such cases so the authors should check that the naming is the same in manuscript and .xls files.

We have now corrected this in the manuscript (p. 11, line 299).

Thank you again for sending us your revised manuscript. We have now heard back from reviewer #3 who was asked to evaluate your study. As you will see below, the reviewer thinks that most issues have been satisfactorily addressed. S/he raises however some remaining issues, all of which refer to text modifications. We would ask you to address these issues in a minor revision.

REFEREE REPORTS

Reviewer #3:

I think the authors has responded well to many of my comments as well as comments from the other reviewers and my points below are directed towards minor corrections of the text to improve the manuscript.

Overall I think the two last result sections are good and have no specific comments on these.

I think the lysate melting curve data set and the reference to the LipMS data has been nicely introduce into the manuscript

The new figure with the distribution of the outer membrane proteins really point to that there is something else going on than only protein melting at the second peak. The fact that many outer membrane proteins do not melt properly (more than 50% residual fold change at highest temperatures) and that the second transition appears to be at very similar temperatures with similar shape, suggest that there are soluble aggregates generated that only partly relate to the intrinsic protein melting behavior. For human cells, care in CETSA/TPP experiments have been taken at higher temperatures where there is a risk of membrane rupture (eg. PMID:23828940). There is a literature on reorganization of the outer membrane of E coli above 55 deg (eg. PMID:3901917) where this peak starts. It is possible that the second peek is contributed by the rupture (phase transition) of the outer membrane, on top of the specific protein unfolding events, leading to soluble fragments/aggregates. The authors should discuss this possibility in relationship to the existing literature and what they see in their data.

In general, the author provide support for the that CETSA/TPP works for a subset of membrane proteins which is very nice but in their previous Nature Method paper, in fact, most of the membrane proteins expected to bind the compounds used were not shifting. It should be noted that also experiments without detergents and the high spin used in the preset work, gives "melting curves" of membrane proteins (still sitting in lipid bilayer vesicles) but they do typically not give shifts. For CETSA/TPP to report on binding events, only the well folded membrane proteins should be solubilized with the detergent, not partly unfolded or unfolded proteins. Due to the complexity of the steps of the method, getting melting curves is therefore not sufficient to support that proteins will shift upon ligand binding. The lack if shifts for many membrane proteins in their previous study suggest that the detergent does often not give perfectly solubilized membrane proteins but instead solubilized aggregates of unfolded or partially unfolded membrane protein. This also relates to my point on localization, which I still feel is a very forced conclusion. I think over-emphasizing this gives a bad impression of the paper, partly due to the membrane protein issues, partly due to that only 3 localization are used for the conclusion (poor counting statistics ...), and that (in my view) the extrema outliers seams to contribute a lot as seen in the figure. In particularly the statement in the abstract "thermostability depending on subcellular location-forming a high-to-low gradient from the cell surface to the cytoplasm" give an impression of overselling a conclusion which is hardly there.

The growth phase study might be fine but there could also be clarifications needed here which I missed in my initial report. CETSA/TPP can be very sensitive to metabolite concentrations in the media when this effects intracellular concentrations. The two growth phases are studied at very different ODs, dilutions, media changes etc which could potentially dramatically change the levels of metabolites in the media. However, in the MM section it is hard to follow how these steps are made and the authors should clarify this so it is possible to appreciate potential changes in

media/metabolite concentrations. They should further clarify in the main text what could be different in the media concentration of the two growth phases and whether this could affect their conclusions (as far as I understand the protocol, it looks as the shifts are due to "flux", i.e. are in the opposite direction to potential media changes and that the authors conclusions are ok).

Overall, the experimental protocols have been significantly improved but the exact brand, pore size, etc of the multi-well filtration membrane used in the CETSA experiment should be described as well as details for how the spin was performed

Some improvements have been made in the discussion on the physical basis for the shifts. I still feel the authors could have done a better job here, in particularly for the metabolic enzymes. Most of these are very well describe with detailed information on biochemistry, structures and ligands. The authors could be distinct on whether a stabilizing regulator/allosteric ligand is known or not (e.g. in a table in the supplement). If not, the stabilization should be due to a substrate (flux) or product (inhibition) interaction. As now written, in spite of that the (magic...) activation statement is deleted, it is not coming out clearly which are even the possibilities for these stabilizations. The authors should make a second effort to lift these discussions.

I think the complex analysis and discussion is very significantly improved. In reference to Tan et al, they should specifically write out that the phenomena is called Thermal Proximity Coaggregation (TPCA). I think it is unfortunate that the authors try and give the impression in their revision letter that they were not aware of the Tan work at their submission. This when the senior author presented in the same section as Chris Tan in a meeting several month before submission (Sep 2017, <https://sites.google.com/view/cetsaworkshop/program-location>), where many groups using CETSA worldwide was present. They might have had similar ideas but they were certainly aware of the Tan work.

Another point on terminology. Although TPP has been used throughout the paper this is all CETSA with MS detection. Several groups therefore call the method MS-CETSA which is better referencing the origin and biophysical principle of the method. However, these groups also acknowledge that it is called TPP by the Heidelberg groups and others. Therefore, when introducing the term TPP, it should also be mentioned that the method is called MS-CETSA.

2nd Revision - authors' response

21st June 2018

Author's second Point by Point response.

Reviewer #3:

I think the authors has responded well to many of my comments as well as comments from the other reviewers and my points below are directed towards minor corrections of the text to improve the manuscript.

Overall I think the two last result sections are good and have no specific comments on these.

I think the lysate melting curve data set and the reference to the LipMS data has been nicely introduce into the manuscript

The new figure with the distribution of the outer membrane proteins really point to that there is something else going on than only protein melting at the second peak. The fact that many outer membrane proteins do not melt properly (more than 50% residual fold change at highest temperatures) and that the second transition appears to be at very similar temperatures with similar shape, suggest that there are soluble aggregates generated that only partly relate to the intrinsic protein melting

behavior. For human cells, care in CETSA/TPP experiments have been taken at higher temperatures where there is a risk of membrane rupture (eg. PMID:23828940). There is a literature on reorganization of the outer membrane of E coli above 55 deg (eg. PMID:3901917) where this peak starts. It is possible that the second peak is contributed by the rupture (phase transition) of the outer membrane, on top of the specific protein unfolding events, leading to soluble fragments/aggregates. The authors should discuss this possibility in relationship to the existing literature and what they see in their data.

The reviewer is right that it is possible that there are changes in the outer membrane in that temperature range. We have added a statement to the results section to bring attention to this.

In general, the author provide support for the that CETSA/TPP works for a subset of membrane proteins which is very nice but in their previous Nature Method paper, in fact, most of the membrane proteins expected to bind the compounds used were not shifting. It should be noted that also experiments without detergents and the high spin used in the preset work, gives "melting curves" of membrane proteins (still sitting in lipid bilayer vesicles) but they do typically not give shifts. For CETSA/TPP to report on binding events, only the well folded membrane proteins should be solubilized with the detergent, not partly unfolded or unfolded proteins. Due to the complexity of the steps of the method, getting melting curves is therefore not sufficient to support that proteins will shift upon ligand binding. The lack of shifts for many membrane proteins in their previous study suggest that the detergent does often not give perfectly solubilized membrane proteins but instead solubilized aggregates of unfolded or partially unfolded membrane protein.

We agree with the reviewer that certain membrane proteins might not show observable shifts, but we do observe some in our study (e.g., AcrB or MdtF in Δ tolC cells). In addition, there are other publications meanwhile showing thermal shifts of membrane proteins detectable by TPP or CETSA, using detergent (PMID: 29851333, PMID: 29551266, PMID: 27669419).

This also relates to my point on localization, which I still feel is a very forced conclusion. I think over-emphasizing this gives a bad impression of the paper, partly due to the membrane protein issues, partly due to that only 3 localization are used for the conclusion (poor counting statistics ...), and that (in my view) the extrema outliers seems to contribute a lot as seen in the figure. In particularly the statement in the abstract "thermostability depending on subcellular location-forming a high-to-low gradient from the cell surface to the cytoplasm" give an impression of overselling a conclusion which is hardly there.

We again politely disagree with the reviewer. First, we would like to point out that we use non-parametric statistical tests that do not suffer from outlier bias. Second, we observe a higher T_m for periplasmic proteins than cytosolic proteins (which are soluble proteins and therefore are not affected by the detergent extraction; the higher T_m is probably due to the presence of disulfide bonds in proteins present in this compartment). Third, multiple complexes of proteins with inner membrane and periplasmic components co-melt, which gives us some confidence on the T_m 's of inner membrane proteins. Fourth, our observation of heat-resistant outer membrane

proteins only confirms previous knowledge that these are particularly resistant to heat-induced denaturation once folded and assembled in the membrane. Therefore, we stand by our conclusions that cytosolic proteins have in general lower T_m , outer membrane proteins are very thermostable, and inner membrane and periplasmic proteins are somewhere in the middle.

The growth phase study might be fine but there could also be clarifications needed here which I missed in my initial report. CETSA/TPP can be very sensitive to metabolite concentrations in the media when this effects intracellular concentrations. The two growth phases are studied at very different ODs, dilutions, media changes etc which could potentially dramatically change the levels of metabolites in the media. However, in the MM section it is hard to follow how these steps are made and the authors should clarify this so it is possible to appreciate potential changes in media/metabolite concentrations. They should further clarify in the main text what could be different in the media concentration of the two growth phases and whether this could affect their conclusions (as far as I understand the protocol, it looks as the shifts are due to "flux", i.e. are in the opposite direction to potential media changes and that the authors conclusions are ok).

As we describe in the methods section (p. 16):

- For the experiments in exponential phase: an overnight culture of cells was diluted 2000-fold into 50 ml of LB medium and cells were grown until OD of 0.1.
- For experiments in stationary phase: an overnight culture of cells was diluted 100-fold into 10 ml of LB medium and cells were grown until OD of 2.

The reason for different dilution factors and culture volumes was to be able to do the experiments in parallel from the same initial overnight culture and harvest the cells after the same time (same number of divisions). The low dilution does not affect the physiology of the cell in stationary phase, and the high dilution is required to have cells in steady-state exponential phase.

As the reviewer suggests, different nutrients will be available in the medium (especially since we utilize a rich medium) when the cells are harvested, and it is precisely these that will induce the (de)stabilization of different metabolic enzymes – either through flux or inhibition (as pointed out by the reviewer below).

Overall, the experimental protocols have been significantly improved but the exact brand, pore size, etc of the multi-well filtration membrane used in the CETSA experiment should be described as well as details for how the spin was performed

The reviewer is right. We have now added those details to the manuscript (p. 16).

Some improvements have been made in the discussion on the physical basis for the shifts. I still feel the authors could have done a better job here, in particularly for the metabolic enzymes. Most of these are very well describe with detailed information on biochemistry, structures and ligands. The authors could be distinct on whether a stabilizing regulator/allosteric ligand is known or not (e.g. in a table in the supplement). If not, the stabilization should be due to a substrate (flux) or product (inhibition) interaction. As now written, in spite of that the (magic...) activation statement is deleted, it is not coming out clearly which are even the possibilities for these stabilizations. The authors should make a second effort to lift these discussions.

Thank you for this comment. We already discuss 11 of 39 hits in depth (including substrates) in the results section – linking back to the relevant literature. We now also add a statement at the beginning of the results section where we explain the we expect the shifts to be caused by flux, inhibition or allosteric regulation (p. 6):

“We tested the impact of growth phase on the thermal stability of the proteome, since we expected differences in protein activity to be reflected in thermal stability (Reinhard et al, 2015; Savitski et al, 2014)—for example, proteins might be stabilized by substrates (indicating flux through the pathway), products (indicating inhibition) or allosteric regulators.”

I think the complex analysis and discussion is very significantly improved. In reference to Tan et al, they should specifically write out that the phenomena is called Thermal Proximity Coaggregation (TPCA). I think it is unfortunate that the authors try and give the impression in their revision letter that they were not aware of the Tan work at their submission. This when the senior author presented in the same section as Chris Tan in a meeting several month before submission (Sep 2017, <https://sites.google.com/view/cetsaworkshop/program-location>), where many groups using CETSA worldwide was present. They might have had similar ideas but they were certainly aware of the Tan work.

This is a peculiar comment. There was obviously no possibility to cite the work from Tan et al before it was published. We used a completely different analysis method in the first submission, previously used by us to assess protein complex co-turnover (Mathieson et al Nature Communications January 2018).

After the work from Tan et al appeared and we got the opportunity to revise the manuscript we switched to the analysis of Tan et al and gave the paper its due credit.

We now also mention Thermal Proximity Coaggregation (TPCA) (p. 7) explicitly as requested by the reviewer, even though the request is a bit odd since the Tan et al was very clearly, visibly, and respectfully cited already.

Another point on terminology. Although TPP has been used throughout the paper this is all CETSA with MS detection. Several groups therefore call the method MS-CETSA which is better referencing the origin and biophysical principle of the method. However, these groups also acknowledge that it is called TPP by the

Heidelberg groups and others. Therefore, when introducing the term TPP, it should also be mentioned that the method is called MS-CETSA.

This is a peculiar comment. We published the first ever combination on mass spectrometry and cellular thermal shift in *Science* in 2014 and called the technology “thermal proteome profiling” which is a widely accepted term (150 hits on google scholar when searching for exact match). We respect that some other people have recently started using a new naming convention, we do not oppose it in anyway and we are very happy that the original name is still mentioned by the groups.

We do however reserve ourselves the right to use the originally introduced name “thermal proteome profiling” and to not mention alternative names. We also already cited the cellular thermal shift work and the relevant mass spectrometry work in our first submission.

Corresponding Author Name: Mikhail M. Savitski and Athanasios Typas

Manuscript Number: MSB-18-8242